



# Modelling Frontal Low-Level Jets and Associated Extreme Wind Power Ramps over the North Sea

Harish Baki[1], Sukanta Basu[2,3], and George Lavidas[4]

[1]Geosciences and Remote Sensing, Faculty of Civil Engineering and Geosciences, Delft University of Technology, Delft, Netherlands
[2]Atmospheric Sciences Research Center, State University of New York, Albany, the United States
[3]Department of Environmental and Sustainable Engineering, University at Albany, Albany, the United States
[4]Marine Renewable Energies Lab, Offshore Engineering, Hydraulic Engineering Department, Faculty of Civil Engineering and Geosciences, Delft University of Technology, Delft, Netherlands

**Correspondence:** Harish Baki (h.baki@tudelft.nl)

**Abstract.** The increasing global demand for wind power underscores the importance of understanding and characterizing extreme ramp events, which are significant fluctuations in wind power generation over short periods, that pose challenges for grid integration. This study focuses on modeling frontal low-level jets (FLLJs) and associated extreme ramp-down events, particularly their impact on wind power production at Belgium offshore wind farms. Using the Weather Research and Forecasting (WRF) model, we analyzed five cases of extreme wind power ramp down events, including in-depth analysis of two cases and generalization of three additional cases. We assessed the sensitivity of various model configurations, including initial and boundary condition (IC/BC) datasets (ERA5 and CERRA), the activation of Fitch wind farm parameterization (WFP), planetary boundary layer (PBL) schemes, and single versus nested-domain configuration. Our findings indicate that CERRA IC/BCs provide a superior representation of atmospheric flow compared to ERA5, resulting in more accurate predictions of ramp timing, intensity, and FLLJ characteristics. The WFP significantly impacts wind power output by modeling turbine interactions and wake effects, leading to slightly lower wind speeds. The scale-aware Shin and Hong PBL scheme yielded a stronger FLLJ core at higher altitudes with a more pronounced jet nose, although wind speeds below 200 m were lower compared to the Mellor-Yamada-Nakanishi-Niio 2.5 scheme. Single-domain configuration proved more effective in simulating wind power ramps, although higher core heights and higher wind speeds below 200 m, resulting in a diffused jet profile. Our analysis highlights that reliable simulation of extreme ramps associated with FLLJ using a single domain configuration could reduce computational costs. Further, the FLLJ and associated extreme ramps can be predicted one day in advance, offering substantial benefits for operational efficiency in wind energy management.

## 1 Introduction

The demand for global wind power is surging as renewable energy is increasingly seen as essential to combat climate change. The European Union aims for 32% renewable energy by 2030 to reduce greenhouse gas emissions by at least 55% (European Union, 2018). Offshore wind farms (OWFs) are pivotal in meeting these goals, despite their higher construction and mainte-




nance costs compared to onshore wind farms (Zheng et al., 2016). This is due to two main factors: the higher capacity factor of OWFs, as offshore winds are generally 25% stronger and can be utilized 2-3 times longer than onshore winds (Tambke et al., 2005), and technological advancements that allow for the use of larger, more efficient turbines (Mehta et al., 2024). By 2019, Europe's OWF capacity reached 22 GW, with 77% located in the North Sea. The EU plans to expand offshore wind capacity to 450 GW by 2050, with 47% (212 GW) in the North Sea, necessitating an annual consenting rate of 8.8 GW (Wind Europe, 2019). This makes the North Sea a critical area for OWF development.

A major challenge for integrating large-scale wind power into the grid is managing extreme ramp events, which are significant fluctuations in wind power generation over short periods (Ueckerdt et al., 2015; Mararakanye and Bekker, 2019). Without large-scale energy storage, power production must match power consumption, a balance maintained by transmission system operators (TSOs). Often, ramp-up events necessitate wind farm curtailment because surplus electricity cannot be dispatched, leading to profit loss for farm owners. At other times, when wind power output suddenly increases, conventional power generation, such as coal and natural gas, must reduce output, a process known as cycling (Veron et al., 2018). This cycling lowers the capacity credit of conventional generation. In a more complex manner, during ramp-down events, when wind power output suddenly drops, the supply from conventional power generators must increase to meet demand. TSOs often purchase these resources in competitive markets at high rates. In critical situations, load shedding becomes necessary due to minimal power output from wind farms. Wind farm owners then incur additional costs when they fail to meet specific loads and quotas (Valldecabres et al., 2020).

These challenges were evident in the case of Great Britain offshore wind farms on 3rd November 2014 (Drew et al., 2017). During that day, wind power output increased by 1.1 GW in 2 hours and 45 minutes (ramp-up), followed by a 1.2 GW decrease in 1 hour and 50 minutes (ramp-down). Analysis of the meteorological conditions has shown that the ramp-up was caused by a trough that formed behind a large weather front, while the ramp-down was caused by the subsidence of the front. This unforecasted event caused a market surplus of 570 MWh at 15:30 due to unexpected generation in the Thames Estuary, and a deficit of 820 MWh at 17:00 due to the sudden drop in generation. The imbalance coincided with high demand, leaving fewer available generation units. Expensive short-term operating reserves were deployed, causing a system buy price spike to £183 per MWh, the 16th highest price of the year. This event underscores the importance of a comprehensive understanding of power ramp events and the atmospheric phenomena that contribute to their occurrence.

Ramp-up events are primarily caused by synoptic-scale weather phenomena, such as cyclones (Lacerda et al., 2017; Drew et al., 2018) and cold fronts (Cutler et al., 2007; Linden et al., 2012; Cheng et al., 2013; Haupt et al., 2014; Marjanovic et al., 2014; Pichault et al., 2021; Pereyra-Castro and Caetano, 2022; Arthur et al., 2020; Veron et al., 2018). These events can also be triggered by mesoscale phenomena, such as thunderstorms (Hawbecker et al., 2017; Cheng et al., 2013; Drew et al., 2018; Pichault et al., 2021) and low-level jets (Freedman et al., 2008). Conversely, ramp-down events occur due to a decrease in wind speed, often triggered by the relaxation of cold fronts (Drew et al., 2018; Zhao et al., 2019; Dalton et al., 2019) and the passage of warm fronts (Cheneka et al., 2021). Practically, ramp-down events pose a greater problem than ramp-up events, primarily due to the significant penalties that wind farm owners may incur from short supply. Additionally, ramp-down events can lead to a decrease in system frequency (Ela and Kirby, 2008).



Gallego-Castillo et al. (2015) pointed out the importance of modeling and understanding the causative meteorological phenomena of extreme ramps, which would help improve ramp forecasting frameworks. Our paper focuses on modeling one such weather phenomenon called the frontal low-level jet (FLLJ) associated with cold fronts, which tends to cause extreme wind power ramp-down events, using the Weather Research and Forecasting (WRF) model.

Previous studies, such as Freedman et al. (2008) suggested that a gradual, rather than a sharp, decline in wind speed is more common during cold fronts, making up-ramps more frequent than down-ramps. This is evidenced by a large number of studies focused on ramp-up events associated with cold fronts. However, the FLLJ causes an abrupt wind speed drop that triggers an extreme ramp-down event on a time scale of 15 to 60 minutes, which is a unique flow feature we are reporting in the present study. To the best of our knowledge, we are not aware of peer-reviewed journal articles associating FLLJs with extreme wind power ramp-down events. In fact, there has been a study from our group as a non-peer-reviewed master's thesis work (Dreef, 2019), which conducted some very preliminary cases in Scotland. That prompted a detailed investigation that led to this particular work.

Larger, synoptically driven features have longer time scales and are theoretically more straightforward to model and forecast than local-scale phenomena, which usually require fine-scale information about land-surface conditions and turbulent mixing in the atmosphere (Marjanovic et al., 2014). The FLLJ, being a synoptic-scale phenomenon, is expected to be accurately modeled and forecasted with relative ease. However, atmospheric phenomena occurring within the atmospheric boundary layer (ABL), such as low-level jets, exhibit significant sensitivity to model physics parameterizations, domain configuration, and forcing data (Storm et al., 2009; Nunalee and Basu, 2014; Aird et al., 2021; Tay et al., 2021; Larsén and Fischereit, 2021; Wagner et al., 2019).

As a first step towards better understanding, we investigate the modeling efforts of extreme wind ramp-down events associated with FLLJs, and their impact on wind power production, using the WRF model (Skamarock et al., 2019). In the mesoscale modeling community, it is well known that the forcing data significantly influence the simulation accuracy (Carvalho et al., 2014; de Linaje et al., 2019; Hahmann et al., 2020). Nonetheless, the fifth generation ECMWF atmospheric reanalysis (ERA5) (Hersbach et al., 2020), existing at a ~32 km resolution, has been the favorite forcing data in wind resource modeling (Olauson, 2018; Ramon et al., 2019; Gualtieri, 2022). However, a recent development in meteorological reanalysis is the Copernicus Regional Reanalysis for Europe (CERRA) (Schimanke et al., 2021), offering a significantly improved resolution of 5.5 km. Despite the advantages of CERRA, the data have never been used as forcing data in wind resource modeling. Therefore, the present study aims to capitalize on the capabilities of CERRA data as initial and boundary conditions. The improved resolution of CERRA forcing data makes it possible for such modeling with detailed representation of atmospheric processes and less computational resource utilization. In addition, the FLLJ being an ABL phenomenon, the model accuracy will be more sensitive to the use of appropriate planetary boundary layer (PBL) parameterization (Nunalee and Basu, 2014; Gevorgyan, 2018; Vemuri et al., 2022). The WRF model has the capability to parameterize the interaction of wind turbines with atmospheric flow through a wind farm parameterization (WFP). However, in the absence of wind farm specifications, it is necessary to understand to what extent the WFP contributes to the modeling accuracy.





Keeping these modeling parameters in mind, we aim to determine how well the FLLJ and the associated ramp-down event can be modeled and assess their impact on wind power production at the Belgium offshore wind farm. We also seek to evaluate the sensitivity of ramp characteristics in terms of timing and intensity, as well as the FLLJ characteristics in terms of core strength and height, to the choice of forcing data, PBL schemes, WFP, and domain configuration.

The organization of this paper is as follows. Section 2 briefly discusses the meteorological background of the FLLJ. The methods used here are introduced in section 3, including the data, description of two case studies, a diverse set of WRF model configurations, and the simulation setup. The results will be presented in section 4. The conclusions and the future scope will be presented in section 5. The appendix consists of additional information, including the description of three additional case studies identified for modeling, comparison between ERA5 and CERRA forcing data, and illustration of some additional

results.

## 2  Frontal low-level jet: A meteorological perspective

A Low-Level Jet (LLJ) is a local maximum in the vertical wind speed profile, often referred to as a core or sometimes a nose, characterized by a modest (approximately 2-3 m/s) decrease in wind speed above and below the core. LLJs typically occur within the planetary boundary layer (Bonner, 1968; Hallgren et al., 2023). They can be induced by various mechanisms,

including inertial oscillation (Blackadar, 1957; Bonner, 1968), diurnal changes related to surface and terrain characteristics, such as coastal jets (Smedman et al., 1993, 1995; Nunalee and Basu, 2014), large-scale baroclinity influenced by sloping terrain (Xing-Sheng et al., 1983; Gerber et al., 1989), and geostrophic adjustment, as seen in barrier jets (Parish and Oolman, 2010; Li and Chen, 1998).

The Frontal Low-Level Jet (FLLJ) shares similarities with the barrier jet mechanism. Observations by Browning and Har-

rold (1970) first identified the FLLJ while studying air circulation and precipitation growth in cold fronts over the British Isles. They reported strong wind speeds, ranging from 25 to 30 m/s, just ahead of cold frontal surfaces. Subsequent research has confirmed the existence and characteristics of FLLJs (Browning and Pardoe, 1973; Browning, 1974; Browning and Monk, 1982; Browning, 1986; Browning et al., 1998; Orlanski and Ross, 1977; Thorpe and Clough, 1991; Kotroni and Lagouvardos, 1993; Uccellini and Johnson, 1979; Brill et al., 1985; Dudhia, 1993; Chen et al., 1994; Wakimoto and Murphey, 2008; Demirdjian

et al., 2020; Tay et al., 2021). The key characteristics of the FLLJ are as follows:

- It is a band of strong winds with velocities ranging from 25 to 30 m/s, typically located within the 900-850 mb pressure level, and forms ahead of a cold frontal surface.

- It is a synoptic-scale phenomenon that extends several hundred kilometers ahead of the frontal surface.

- The jet's acceleration is driven by the isallobaric term, with Coriolis torque and advective tendency terms contributing
to its propagation perpendicular to the FLLJ.

- An important feature of the FLLJ is the abrupt wind speed drop immediately following the jet, which is accompanied by a significant change in wind direction.





The intense wind speed during the FLLJ, combined with the drastic decrease in wind speed that follows, results in extreme ramp-down events.

Despite its well-documented characteristics, there is a notable lack of modeling studies focused on the influence of FLLJs and associated extreme ramp-down events on wind power production. While the basic characteristics of FLLJs are known, detailed knowledge regarding their impact on extreme ramps and wind energy production remains sparse in the literature.

## 3 Methods

### 3.1 Data and description of cases

During January 2016 to January 2017, the Belgium offshore wind farm cluster (henceforth BOWFC) consisted of three operational projects: the C-Power, Northwind, and Belwind-I (Li et al., 2021), which had a combined capacity of 712MW. The aggregated power production data from these wind farms is available at a sampling rate of every 15 minutes, which has been quality-controlled by Elia to handle any missing data.

    During the study period, FUGRO deployed two SEAWATCH wind LiDAR buoys at LOT1 (latitude: $51°42.414'$ and 135 longitude:$3°02.0771'$) and LOT2 (latitude: $51°38.778'$ and longitude: $2°57.0846'$) near BOWFC, as part of a metocean measurement campaign for proposed wind farms near the Dutch coast. The LiDARs measure wind speed and direction at 30 m height and from 40 m up to 200 m at 20 m intervals and record the data at a sampling rate of 10 minutes. The two LiDARs have recorded 100% data during the selected weather events, and the data has been quality controlled and released to the public by https:\www.TNO.nl.

As mentioned earlier, wind speed maximum (with a jet core at a height of 900-850 mb pressure) during the FLLJ and drop right after the frontal passage (Browning and Pardoe, 1973) leads to a severe ramp event. Since the lidar observed wind speeds span only up to 200 m altitude, it is impractical to identify jet nose within this altitude. Consequently, we examined the weather maps provided by the United Kingdom Meteorological Office (UKMO), in conjunction with the Belgium offshore wind farm's measured power production, to identity possible cases such that the frontal passages coincide with extreme ramps in measured 145 wind power. Based on the criteria, five cases are identified, out of which, two are the primary focus, while the additional three cases are used for generalization. A description of the two cases is provided in the following section, while the description of the additional cases is provided in the Appendix.

### 3.1.1 Case 1

On 22nd February 2016, BOWFC experienced a 54% drop in it's measured wind power within 1 hour, beginning from 0630 150 UTC, as shown in Figure 1(top panel). The farm is seen producing a maximum wind power of 620 MW, for more than 12 hours period before experiencing the ramp down event, suggesting the occurrence of peak wind speed during this period. Coinciding, the synoptic weather maps at 0000 UTC on 22nd and 23rd of February (bottom left and right panels of Figure 1, respectively) clearly illustrating a cold front (dark line with triangles pointing the direction of frontal movement) overpassing the wind farm.



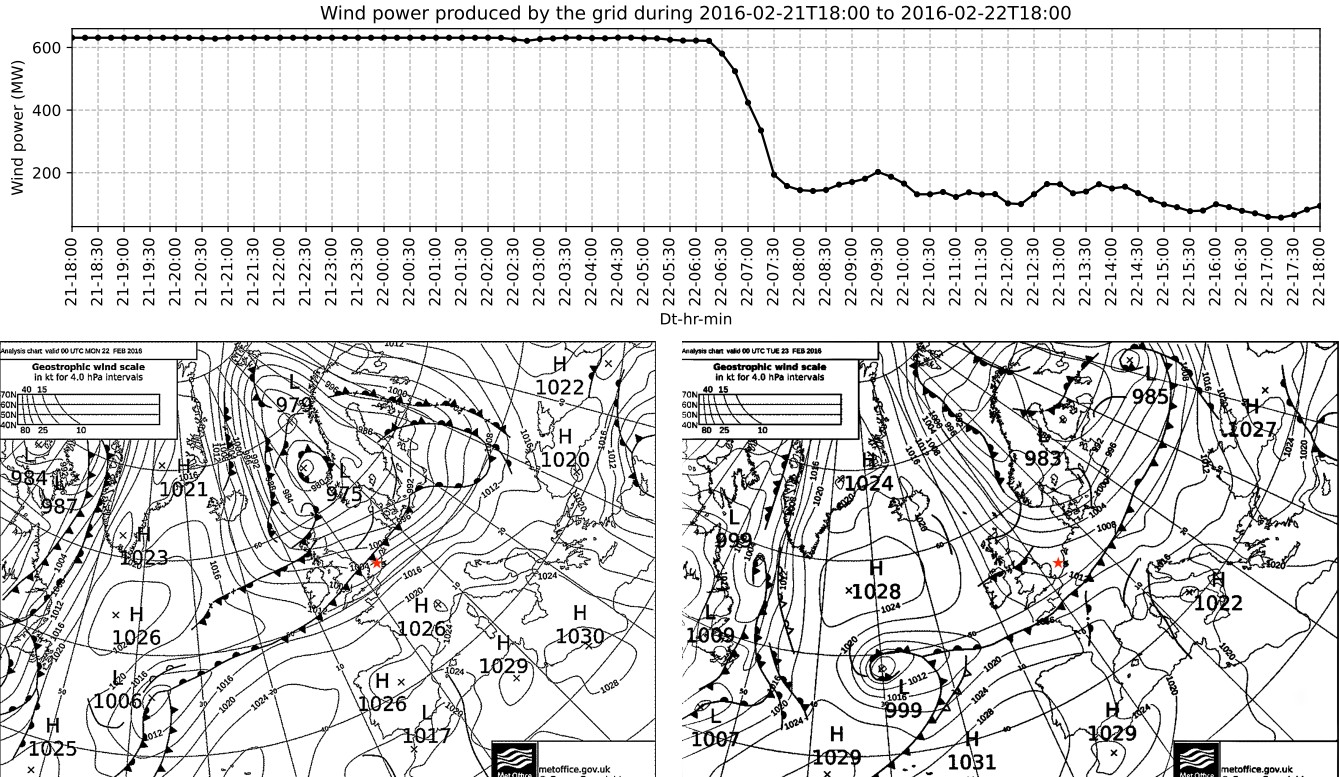

**Figure 1.** The top panel shows the time series of wind power produced by the Belgian offshore wind farms, from 1800 UTC on the 21st to 1800 UTC on the 22nd of February 2016, which depicts the extreme power ramp. The total capacity of the wind farms is 712 MW. Source: https://www.elia.be. Synoptic weather maps on 0000 UTC on 22nd (bottom left panel) and 23rd February 2016 (bottom right panel), illustrating a cold front (represented by black line with filled triangles, pointing towards the frontal movement) passing over the Belgium offshore wind farm (illustrated with red star). Source: https://www.wetterzentrale.de/.

### 3.1.2 Case 2

Similarly to the above, on March 4th, 2016, BOWFC experienced a 88% drop in its measured wind power within 1 hour, beginning from 0700 UTC, as shown in Figure 2(top panel). The farm has been producing power more than 620 MW for four hours before experiencing the severe ramp-down event. The synoptic weather charts at 0000 UTC on 4th and 5th of March (bottom left and right panels of Figure 2, respectively) show that a cold front overpassed the wind farm during this period.

### 3.2 Model configuration

In this study, the WRF model (version 4.4) is utilised to simulate the identified cases. For the sensitivity analysis, we chose to vary the initial and boundary conditions, domain configuration, planetary boundary layer (PBL) schemes, and activation of wind farm parameterization. In total, five hindcast sensitivity experiments and one forecast experiment are designed, which are





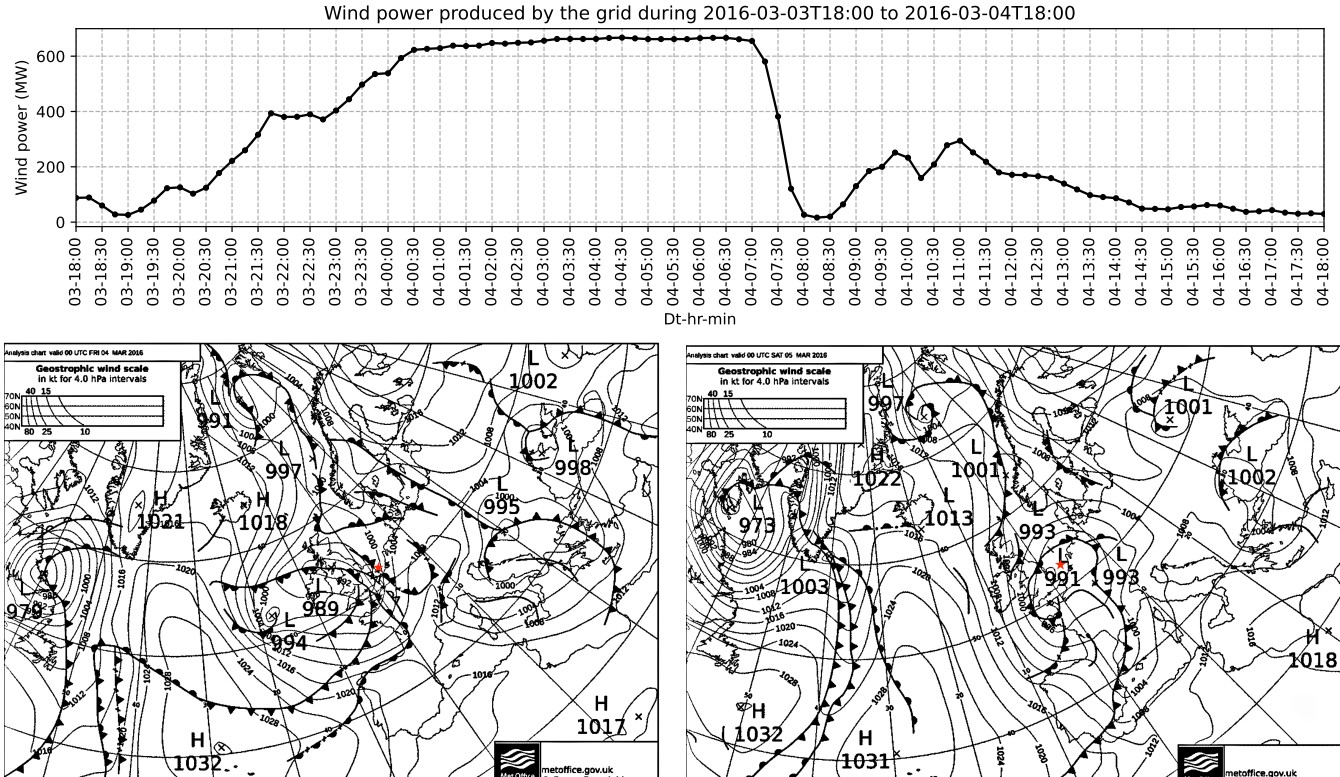

**Figure 2.** Same as Figure 1, but for case 2. The top panel shows the time series of wind power produced by the Belgian offshore wind farms, from 1800 UTC on the 3rd to 1800 UTC on the 4th of March 2016, which depicts the extreme power ramp. Synoptic weather maps on 0000 UTC of 4th March 2016 (bottom left panel) and 0000 UTC of 5th March 2016 (bottom right panel) illustrating a cold front passing over the Belgium offshore wind farm (illustrated with red star).

summarized in Table 1. The innermost domain of all the WRF runs is illustrated in Figure 3, which is consistently maintained across the runs.

### 3.2.1 Initial and boundary conditions

We have explored two options for the initial and boundary conditions (IC/BCs), ERA5 and the CERRA. ERA5 has been released progressively since 2017, and soon after its introduction, it became the preferred reanalysis dataset in the wind power meteorology community. Several studies, like Olauson (2018); Ramon et al. (2019); Gualtieri (2022), and many others, discuss its superior accuracy, lower uncertainty, and higher reliability compared to other global reanalysis datasets. The dataset is available at an hourly resolution and a horizontal grid spacing of about 30km. In this study, the ERA5-3d1kmMYFP runs utilize the ERA5 reanalysis at hourly update frequency as forcing data.



**Table 1.** Overview of the WRF model simulations.

| WRF run | IC/BC (frequency) | Grid size, km | Grid points | PBL scheme | Wind farm parameterization |
|---|---|---|---|---|---|
| ERA5-3d1kmMYFP | ERA5 (1hr) | 9, 3, 1 | 211 × 187, 244 × 244, 301 × 301 | MYNN 2.5 | on |
| CERRA-2d1kmMYFP | CERRA (3hr) | 3, 1 | 181 × 181, 301 × 301 | MYNN 2.5 | on |
| CERRA-2d1kmMYnoFP | CERRA (3hr) | 3, 1 | 181 × 181, 301 × 301 | MYNN 2.5 | off |
| CERRA2d1km-SH | CERRA (3hr) | 3, 1 | 181 × 181, 301 × 301 | SH | – |
| CERRA-1d1kmMYFP | CERRA (3hr) | 1 | 301 × 301 | MYNN 2.5 | on |
| GFS-3d1kmMYFP | GFS (3hr) | 9, 3, 1 | 211 × 187, 244 × 244, 301 × 301 | MYNN 2.5 | on |

CERRA provides a high-resolution pan-European reanalysis with a 5.5 km horizontal resolution and 106 vertical levels, covering Europe, Northern Africa, and Southeastern parts of Greenland. The dataset is essentially downscaled from the global ERA5 reanalysis. Unlike the ERA5 data, CERRA provides analysis every three hours. Despite the advantages of CERRA, the
data have never been used as driving data for simulations and not been incorporated into the WRF model configuration. Thus, we aim to capitalize the fine-scale resolution of CERRA in simulating the extreme ramps associated with the FLLJ events. In doing so, a novel hybrid CERRA-ERA5-based WRF simulation strategy has been developed in this study as follows.

In order to successfully incorporate new forcing data into the WRF model, a new Vtable file needs to be constructed according to the data specifications. Thorough investigation, it was found that all surface and upper-level meteorological variables
needed for the model simulations exist in the CERRA analysis, except soil moisture and soil temperature. Additionally, the WRF model recognizes U and V components of winds, so the 10m wind speed and direction from the CERRA data are transformed into 10m U and V components. A new Vtable corresponding to the CERRA data is created according to the information provided in the downloaded data files. The CERRA analysis data is ungribbed first using the created Vtable file, while the soil moisture and soil temperature variables from ERA5 reanalysis are ungribbed later using a different Vtable file. After this step,



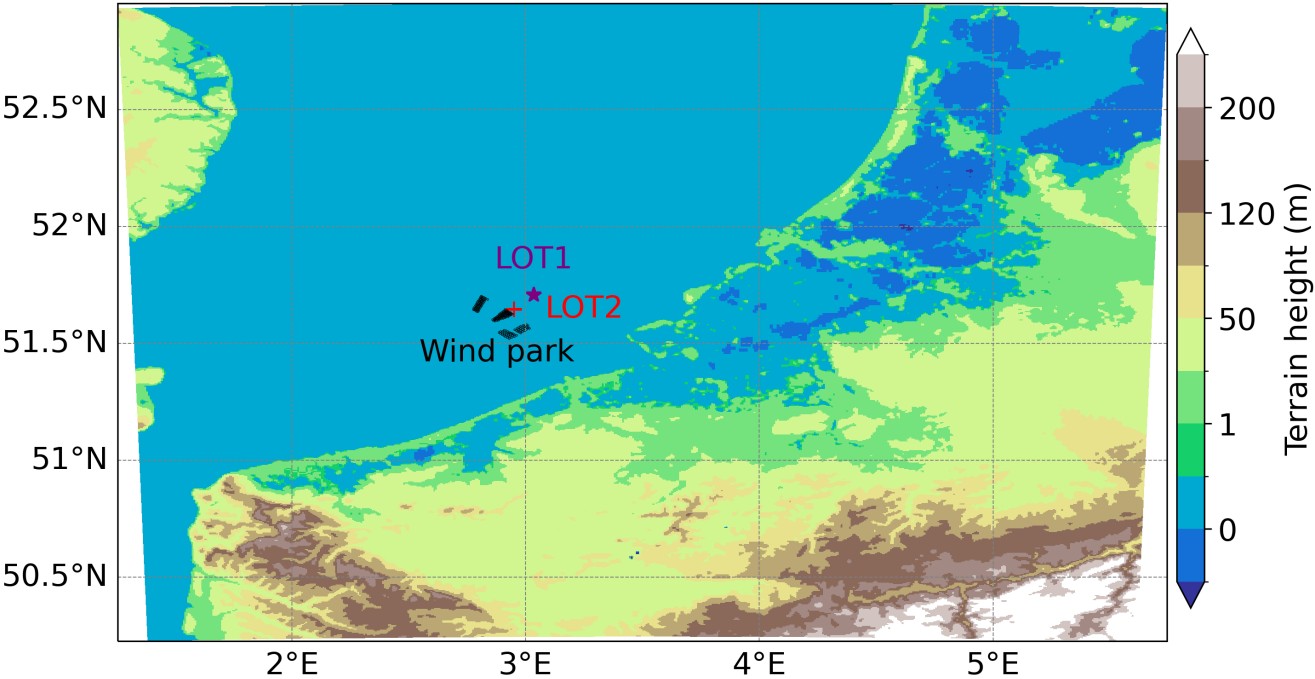

**Figure 3.** A topographical overview of the WRF model simulation domain, encompassing the interested region, consisting of the Belgium offshore wind farm (illustrated with dark filled circles), and two buoy observational locations; LOT1 (illustrated with purple star) and LOT2 (illustrated with red plus). This domain is the innermost domain, that is D03 in ERA5-3d1kmMYFP; D02 in CERRA-2d1kmMYFP, CERRA-2d1kmMYnoFP, and CERRA-2d1kmSH; and D01 in CERRA-1d1kmMYFP simulations.

the final metfiles are created, incorporating all the required meteorological variables for the WRF simulations. Except the ERA5-3d1kmMYFP, the remaining WRF runs utilize the CERRA analysis at three-hourly update frequency as forcing data.

The sensitivity to driving data is assessed through the WRF runs ERA5-3d1kmMYFP and CERRA-2d1kmMYFP. Since, ERA5 reanalysis exist at  32 km resolution, we opted for three one-way nested domains for ERA5-3d1kmMYFP runs, consisting horizontal grid spacing of 9 km, 3 km, and 1 km, in the domains d01, d02, and d03, respectively. On the other hand, 190  the CERRA anaysis exist at 5.5 km resolution, thus we chose two one-way nested domains for CERRA-2d1kmMYFP runs, consisting horizontal grid spacing of 3 km and 1 km, in the domains d01 and d02, respectively.

### 3.2.2  Wind farm parameterization

The wind farm parameterization (WFP) by Fitch et al. (2012) represents the effects of the wind turbines as a drag-induced energy sink and increased turbulence in the vertical levels containing the rotor disk. The Fitch parameterization assumes that a 195  fraction of the total energy flowing through the wind farm is used for power production (based on the turbine power coefficient), and the rest is converted into turbulent kinetic energy (determined by the turbine thrust coefficient). During the study period,





BOWFC consisted of three Belgian offshore wind farms: C-Power, Northwind, and Belwind I, populated with five different types of wind turbines. The wind farms are illustrated as dark fileld circles in Figure 3. More details about each wind turbine, including the power curve, thrust curve, and their corresponding sources, can be found in Li et al. (2021).

The Fitch WFP is well-known for simulating turbine effects, thus it is expected that activating this scheme would result in better wind speed and power simulations. However, the parameterization is only coupled with the MYNN2.5 PBL scheme. To understand the significance of this WFP scheme, we conducted the WRF run CERRA-2d1kmMYnoFP with the WFP turned off, while turned on in the remaining runs with the MYNN2.5 PBL scheme. The sensitivity to Fitch WFP is assessed through the WRF runs CERRA-2d1kmMYFP and CERRA-2d1kmMYnoFP.

### 3.2.3   Planetary boundary layer scheme

Previous studies have emphasized the critical role of PBL schemes in accurately representing wind interactions and turbulence in the lower atmosphere, specifically at the wind turbine hub height (Nunalee and Basu, 2014; Gevorgyan, 2018; Vemuri et al., 2022). Therefore, we chose two different PBL schemes, namely MYNN2.5, which has been employed in many wind energy application studies (Jiménez et al., 2015; Jahn et al., 2017; Li et al., 2021), and Shin Hong (SH) scheme (Shin and Hong, 210   2015), which is a scale-aware scheme, adopted from the studies of Vemuri et al. (2022). The sensitivity to PBL scheme is assessed through the WRF runs CERRA-2d1kmMYnoFP (without Fitch parameterization) and CERRA-2d1kmSH (inherently no WFP).

### 3.2.4   Domain configuration

Since the CERRA analysis exist at 5.5 km resolution, which makes it possible to configure the model domain at kilometer 215   resolution without the need for nesting. Thus, to examine the sensitivity to domain configuration, we configured the CERRA-1d1kmMYFP run with a single domain at 1 km resolution.

The sensitivity to domain configuration is assessed through the WRF runs CERRA-2d1kmMYFP and CERRA-1d1kmMYFP.

### 3.3   Simulation setup

In this study, the MYNN surface layer scheme is used in combination with the MYNN2.5 PBL scheme, while the Revised 220   MM5 surface layer scheme is adopted for the experiments with the Shin-Hong PBL scheme. The remaining physics schemes, including WRF single moment 5-class scheme for microphysics, RRTMG for shortwave and longwave radiation, Unified NOAH for land surface physics, and Kain-Fritsch for cumulus physics, are adopted from the studies of Li et al. (2021). The simulations are conducted with a total of 51 vertical levels, spanning from approximately 8 meters above the surface to around 16 kilometers high, with non-uniform grid spacing. The lowest 1 kilometer of the model atmosphere comprises 18 levels.

Event 1 is simulated from 1200 UTC on 21st February 2016 to 1800 UTC on 22nd February 2016, while Event 2 is simulated from 1200 UTC on 3rd March 2016 to 1800 UTC on 4th March 2016. The simulations run for a total of 30 hours, with the initial 6 hours considered as a spin-up period. Output variables, namely wind speed and direction are recorded at 5-minute intervals.





If the WFP is activated, the WRF simulation can directly provide wind power production data. For simulations without WFP, wind power production is calculated based on King et al. (2014), which utilizes the turbine power curves and the wind speed
at the hub height.

### 3.4 Ramp statistics

In the context of power ramp analysis, we adopt a methodology akin to Drew et al. (2018). A percentage drop in wind power with respect to the rated power is used to capture the intensity and timing of the ramp event, which is computed as,

$$\%\Delta P = \frac{P_{t+\Delta t} - P_t}{P^*} \times 100. \tag{1}$$

Here, $P^*$ is the rated grid power and $P_t$ is the simulated power at time instance $t$. The time scales $\Delta t$ are taken at 15 min, 30 min, and 1 hrs, to quantify the ramp rates.

### 4 Results

#### 4.1 Wind power time-series comparison

The time-series data presented in Figure 4 compares wind power outputs from five WRF simulations to the actual production
of the wind farm for two cases. In case 1, the wind farm experienced a significant decrease in power output starting at 0630 UTC, while in case 2, this decline commenced at 0700 UTC, as detailed in the case descriptions.

During the pre-ramp period (from starting to the ramp occurrence), the wind farm produced a consistent maximum power output, which is attributed to the peak winds associated with the FLLJ. Interestingly, the power output is slightly below the rated capacity of 712 MW. This shortfall is likely due to some turbines being curtailed to prevent damage during high wind speeds
or not being in operation. On the other hand, all the WRF model simulations produced power output at the rated capacity, implying that the FLLJ was simulated well. All the simulations produced substantial power drops during the ramp period in both cases, albeit with timing and amplitude discrepancies.

The ramp statistics are computed using Equation 1 and are listed in Table 2 for both cases. In case 1, the wind farm's power output decreased by 19.9% within 15 minutes from 0715 UTC, by 32.3% within 30 minutes from 0700 UTC, and by
54.3% within 1 hour from 0630 UTC. However, the WRF simulations tend to overestimate the intensity of these power drops. Despite overestimating the intensity of the ramp event, the CERRA-1d1kmMYFP simulation well matched the ramp timing. This discrepancy is attributed to the rated power production prior to the onset of the power ramp.

In case 2, the wind farm experienced power drops of 36.7%, 64.6%, and 88.2% within 15, 30, and 60 minutes from 0730 UTC, 0715 UTC, and 0700 UTC, respectively. In contrast to case 1, all the WRF simulations underestimated the ramp inten-
sities across the time scales. Among all, only the CERRA-1d1kmMYFP simulation closely reproduced both the timing and intensity of the power drop, with a 69.5% decrease within 1 hour, matching the observed timing of 0700 UTC.



**Figure 4.** A comparison of wind power time series from the five WRF model simulations and the Belgium offshore wind farm production. Top panel: during 0100 UTC to 1200 UTC on 21st of February, 2016, for case 1. Bottom panel: during 0200 UTC to 1300 UTC on 4th of March, 2016. The ramp period (1 hour) is shaded in grey.





**Table 2.** Overview of the ramp statistics in wind power in terms of intensity (%) and timing (in parenthesis), at three time-scales: 15, 30, and 60 minutes, obtained from the wind power produced by the Belgium offshore wind farm and the five WRF simulations, during case 1 and case 2. The ramp statistics from measured wind power are presented in bold text.

| Model | Case 1 | | | Case 2 | | |
|---|---|---|---|---|---|---|
| | 15min | 30min | 1hr | 15min | 30min | 1hr |
| **Measured** | **19.9 (07:15)** | **32.3 (07:00)** | **54.3 (06:30)** | **36.7 (07:30)** | **64.6 (07:15)** | **88.2 (07:00)** |
| **ERA5-3d1kmMYFP** | 30.6 (05:00) | 51.2 (04:45) | 69.8 (04:30) | 23.9 (08:45) | 38.1 (08:45) | 52.9 (08:45) |
| **CERRA-2d1kmMYFP** | 29.7 (06:00) | 48.0 (05:45) | 78.4 (05:15) | 19.0 (07:45) | 35.3 (07:45) | 54.1 (07:45) |
| **CERRA-2d1kmMYnoFP** | 21.1 (06:00) | 38.0 (05:45) | 56.4 (05:15) | 16.9 (08:45) | 33.7 (08:30) | 46.8 (08:30) |
| **CERRA-2d1kmSH** | 27.2 (06:15) | 43.0 (06:00) | 67.0 (05:30) | 16.3 (08:00) | 27.9 (07:45) | 42.7 (07:45) |
| **CERRA-1d1kmMYFP** | 28.7 (06:45) | 45.0 (06:30) | 62.8 (06:30) | 35.8 (07:30) | 45.8 (07:15) | 69.5 (07:00) |

Comparing the ERA5-3d1kmMYFP and CERRA-2d1kmMYFP simulations, the ERA5 IC/BCs tend to produce ramps at significantly different times, either earlier or later, although their amplitudes are similar to those of the CERRA IC/BCs.

In the comparison between CERRA-2d1kmMYFP and CERRA-2d1kmMYnoFP, the Fitch WFP consistently simulates
lower power output than the no-FP run. This discrepancy is attributed to the interaction of wind turbines with the flow and the generation of wind wakes. To investigate further, 100 m level wind speed spatial plots from both simulations were examined at various time points for cases 1 and 2, as shown in Figures D1 to D4 in Appendix. The Fitch WFP clearly produced wakes, indicated by a wind speed deficit of 2 m/s along the wind direction. These wakes extend much farther during FLLJ periods. Due to wake generation, the wind speed simulated by the CERRA-2d1kmMYFP run is slower than in its counterpart.
Aside from the vicinity of the wind farm, wind speeds in both runs are almost identical elsewhere.

When comparing CERRA-2d1kmMYnoFP and CERRA-2d1kmSH, the wind power time series are largely similar across cases. The scale-aware SH scheme better captures ramp timing, while the MYNN 2.5 scheme marginally better captures ramp intensity. However, these differences are minimal.

In the comparison between CERRA-2d1kmMYFP and CERRA-1d1kmMYFP, the domain configuration, such as nesting
two domains versus a single domain, significantly influences power output. The single-domain configuration in both cases reproduced the power time series more accurately, matching the observed data, unlike its counterpart. Additionally, ramp timing and intensities are better captured by the single-domain setup.

Overall, there is clear evidence of sensitivity in ramp timing and amplitude with respect to the modeling parameters, with domain configuration and the WFP being the most influential factors.

### 4.2 Wind speed and direction time-serires comparison

The wind speed time-series at different vertical levels from lidar observations at the LOT2 location for cases 1 and 2 are illustrated in Figure 5 (top panel). In both cases, a significant drop in wind speed is clearly evident. What distinguishes the two



cases is that during the pre-ramp period, there are significant wind speed shears within the 30-200 m layer in case 1, whereas they are only slightly noticeable in case 2. Furthermore, the wind speed drop around 0600 UTC in case 1 is drastic, whereas
in case 2, it occurs more gradually from 0600 to 0700 UTC. Nonetheless, in both cases, the wind speed drops at all vertical levels, reaching as low as 6 m/s from as high as 20 m/s in case 1, and 5 m/s from 17 m/s in case 2. Comparing the wind speed time-series with the wind power data, it becomes evident that the ramp timing is one hour ahead in wind speed. This can be explained by the movement of the front and spatial distribution of wind farms, which cannot experience the ramp all at once, whereas lidar observations are measured at a single point.

Similarly, the wind direction time-series from lidar observations at the LOT2 location for cases 1 and 2 are illustrated in Figure 5 (top panel). Significant wind directional shifts, approximately $100°$ in case 1 and $130°$ in case 2, are evident in both cases. These shifts align with the wind speed ramp timings, indicating that the wind speed ramp was accompanied by sharp wind directional changes. Unlike the wind speed shears, there are no visible wind veers between the 30-200 m layer, except for occasional fluctuations.

The WRF-simulated wind speed and direction time-series from the five runs are illustrated in Figures 5 to 6 (2nd to last panel). Comparing the ERA5-3d1kmMYFP and CERRA-2d1kmMYFP, the simulations forced with CERRA reanalysis were able to reproduce the drastic wind speed drop and wind directional shift in both cases, albeit with considerable timing errors, compared to the observations. Similarly, the simulations forced with ERA5 also reproduced a strong wind speed drop and directional shift in case 1. However, in case 2, the ramp is more gradual, spanning several hours, with a strong directional shift
visible three hours later than observed.

The reason for these discrepancies is examined through 100 m wind speed time-series obtained from the original forcing datasets, as shown in Figure B1. ERA5 shows a wind speed drop of 3 m/s in case 1 and 4 m/s in case 2 within one hour, whereas CERRA shows a drop of 9 m/s in case 1 and 4.5 m/s in case 2. From these observations, it is deduced that the coarser representation of the atmosphere in the ERA5 forcing data led to poor dynamical downscaling through the WRF model. In
contrast, the better representation in CERRA resulted in more intense wind speed ramp simulations.

Comparing the CERRA-2d1kmMYFP and CERRA-2d1kmMYnoFP, the Fitch WFP simulated winds are 0.5-1 m/s lower than the counterpart. This is due to the generation of wakes by the WFP, which reduces wind speed within the vicinity of the wind farm, as discussed earlier. Apart from the reduced wind speed, the wind speed and directional time-series are identical to one another, with little to no variations.

Comparing the CERRA-2d1kmMYnoFP and CERRA-2d1kmSH, the wind speed and directional time series show some degree of similarity during the pre-ramp and ramp periods. However, the scale-aware SH scheme simulated winds across the cases are consistently lower than the counterpart and also closely align with the observations. In addition, the wind speed ramp in case 2 is abrupt from the MYNN 2.5 scheme, while the SH scheme simulated an initial sharp ramp followed by a gradual reduction in winds to 5 m/s. These discrepancies suggest that the PBL scheme has some degree of influence on the wind speed
and direction simulation.

Comparing the CERRA-2d1kmMYFP and CERRA-1d1kmMYFP, the wind speed and direction produced by the single domain configuration are consistently lower than those from the nested domain configuration. In fact, the wind speed time



series during the pre-ramp period from the single domain configuration are very close to the observed ones, with an intermittent wind speed drop in case 1 and a gradual rise in wind speed during case 2. The CERRA-1d1kmMYFP simulation is particularly interesting, as it perfectly captured the ramp timing in both cases. However, in case 1, it reproduced a secondary gradual down-ramp, which is not seen in the observations, nor produced by the nested domain configuration.

Since downscaling is largely governed by the forcing data, the nested domain configuration simulations will considerably modify the initial and boundary conditions (IC/BCs) due to telescopic refining, while the single-domain simulations remain closer to the original IC/BCs. Thus, the better accuracy observed in the single-domain configuration simulation is attributed to the more accurately resolved atmospheric features present in the CERRA IC/BCs.

## 4.3 Wind profiles

To identify the FLLJ and its characteristics in terms of core strength and height, we need wind speed profiles spanning the entire boundary layer. Unfortunately, lidar observations only measure wind speed up to 200 m, which is inadequate for characterizing FLLJs. As an alternative, we utilized the WRF-simulated wind profiles from five runs for the analysis. Knowing that the jet occurs during the pre-ramp period, we extracted the hourly averaged wind speed profiles during different time instances at the LOT2 location, illustrated in Figure 7 for both cases. Apart from the wind speed profiles, the time-height cross-sections of wind speed at the LOT2 location are also illustrated in Figure C1.

Although there is no universally accepted definition of an LLJ, in this study, we used the definition from Hallgren et al. (2023): an increase in horizontal wind speed of at least 1 m/s and 10% of the core speed below the jet core (jet nose) and simultaneously a decrease of 1 m/s and 10% above the core. The LLJ core height is marked with a black solid line in Figure C1.

Comparing the ERA5-3d1kmMYFP and CERRA-2d1kmMYFP, in case 1, the ERA5 IC/BCs produced diffused wind profiles at all times, with no defined jet nose identified. On the other hand, the CERRA IC/BCs produced a strong FLLJ core at 400 m altitude. In case 2, both IC/BCs produced a defined jet nose, with the CERRA-forced profiles being more diffuse than the ERA5-forced profiles. This suggests that the vertical profiles from the WRF runs are greatly influenced by the forcing data.

To clarify the source of this discrepancy, we investigated the wind speed profiles from the forcing data (metfiles of ERA5 and CERRA simulations' parent domain), as shown in Figure B2, during 0500 UTC for case 1 and 0600 UTC for case 2. Interestingly, the ERA5 forcing data in case 1 resembles an LLJ profile, with a core height at 700 m. However, the ERA5-3d1kmMYFP run failed to reproduce this LLJ profile. Conversely, the CERRA forcing data in case 2 does not resemble an LLJ profile, but the CERRA-2d1kmMYFP run was able to produce an LLJ profile.

Comparing the CERRA-2d1kmMYFP and CERRA-2d1kmMYnoFP, the WFP does not seem to have any significant influence on the vertical wind speed profiles, except for the occasional lower jet core height, which can be attributed to the reduced wind speed. Comparing the CERRA-2d1kmMYnoFP and CERRA-2d1kmSH, the scale-aware SH scheme results in a stronger FLLJ core at a higher altitude compared to the MYNN 2.5 scheme across the cases. Additionally, the FLLJ nose is more defined and pronounced in the scale-aware PBL scheme, suggesting a more accurate representation of the jet characteristics.

Comparing the CERRA-2d1kmMYFP and CERRA-1d1kmMYFP, the single domain configuration notably produced double peaks in case 1, which seem to stem from the forcing data (as observed in the CERRA profiles from Figure B2), indicating that



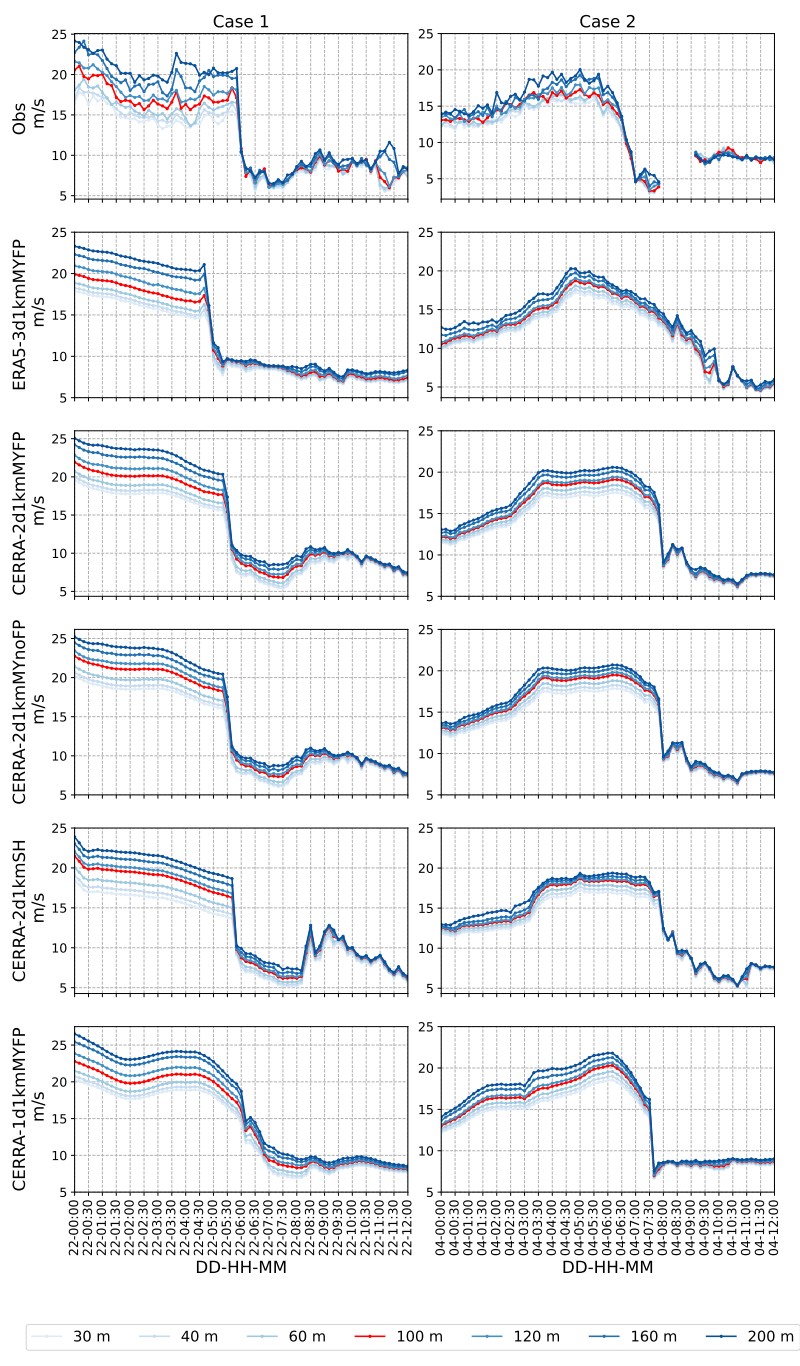

**Figure 5.** A comparison of wind speed time series at seven vertical levels from lidar observations at LOT2 (1st row) and the five WRF model simulations (consecutive rows). Left panel: during 0100 UTC to 1200 UTC on 21st of February, 2016, for case 1. Right panel: during 0200 UTC to 1300 UTC on 4th of March, 2016, for case 2.

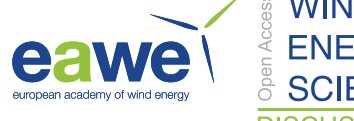

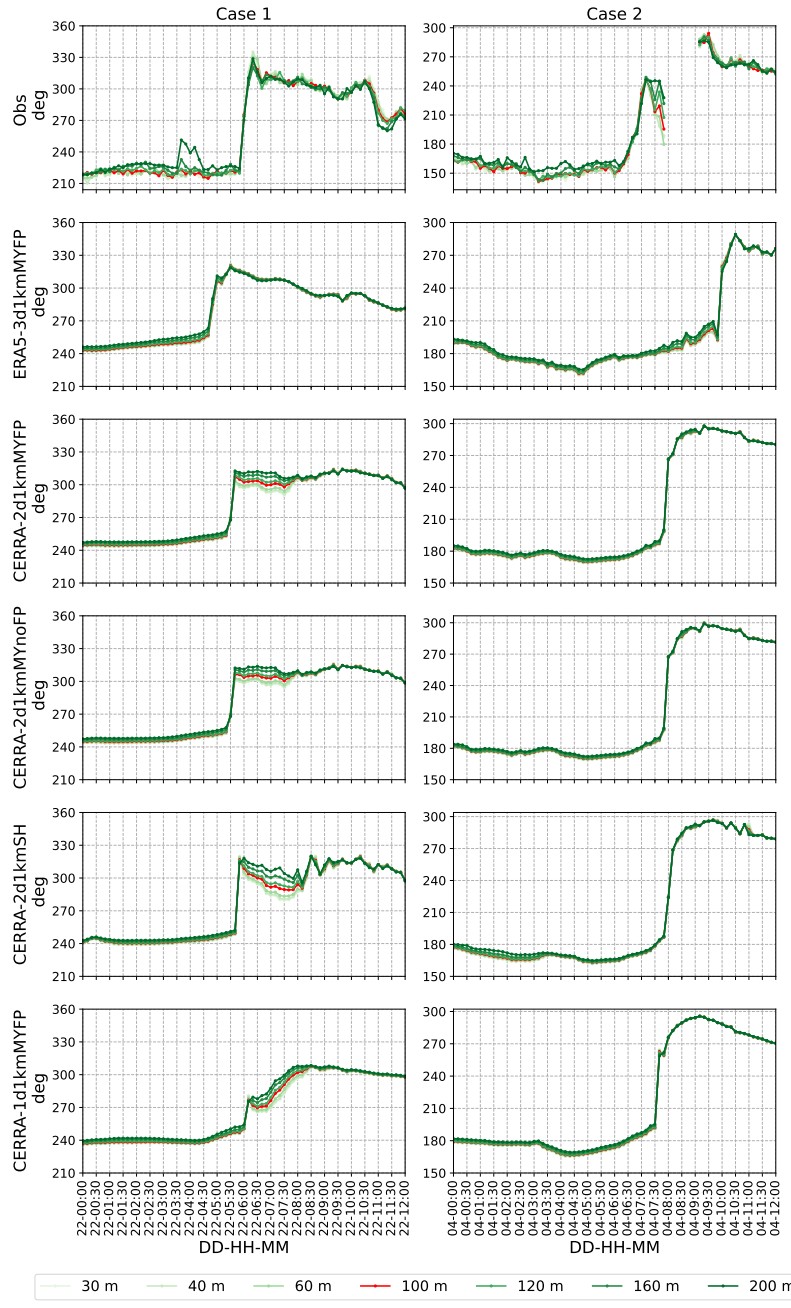

**Figure 6.** A comparison of wind directiion time series at seven vertical levels from lidar observations at LOT2 (1st row) and the five WRF model simulations (consecutive rows). Left panel: during 0100 UTC to 1200 UTC on 21st of February, 2016, for case 1. Right panel: during 0200 UTC to 1300 UTC on 4th of March, 2016, for case 2.





the downscaling is largely governed by the forcing data here. In both cases, the single domain configuration produced jet cores at higher altitudes with higher magnitudes than the double nested domain configuration.

Overall, all simulations were able to produce some degree of jet core, but with varying heights and strengths. Interestingly, the wind profiles are different across the cases, signifying the diversity in characteristics of the two case studies. Since the profiles are derived from simulations, it is difficult to determine which one is the most accurate. However, it is clear that an FLLJ exists in both cases, with a wind speed maximum (core strength) of 25 m/s. A clear jet core is located around 700 m in case 2, but there is ambiguity in case 1, where the jet core could be around 400 m or 800 m. The differences in the vertical wind speed profiles across the simulations suggest that while FLLJs are a common feature, their precise characteristics can vary significantly depending on the model configuration and resolution.

### 4.4 Synoptic characteristics of FLLJ and the associated ramp

To examine the modeling capability of the WRF model in simulating FLLJ and understanding their synoptic characteristics, we conducted surface and cross-sectional analyses of wind speed. The analysis was performed at the wind speed ramp time instance over the observational site LOT2. From the wind speed time-series shown in Figure 5, it is evident that the WRF runs produced wind speed ramps at different time instances. Thus, to eliminate any influence of temporal shift on the FLLJ synoptic characteristics, we chose wind speed from each WRF run at the corresponding ramp instance, instead of the lidar observed ramp instance.

Accordingly, 200 m level wind barbs from the five WRF runs are presented in Figure 8 (left panel) for case 1 and Figure 9 (left panel) for case 2. From all the runs, the wind barbs reveal two distinct systematic wind patterns separated by sharp shifts in wind direction. This separation represents the boundary between warm and cold air masses, referred to as the surface cold front (SCF) by Browning and Pardoe (1973). From the WRF runs, the SCF is nearly parallel to the Belgium coast in case 1, while aligning with the 3° longitude in case 2 (except for the ERA5-3d1kmMYFP, in which the SCF seems to be curved).

Winds ahead of the frontal boundary, almost parallel to the SCF, are much stronger than the winds trailing the SCF. From the studies of Browning and Pardoe (1973), such strong winds ahead and parallel to the SCF are indicative of an FLLJ. Conversely, the winds trailing the front, expected to be normal to the SCF, display a slight inclination towards the front in case 1, while remaining perfectly normal in case 2.

Differences in wind speed magnitudes across the WRF runs highlight the influence of modeling parameters. However, the orientation and extent of the FLLJ are consistent across runs, except for ERA5-3d1kmMYFP in case 2. Wind speeds from ERA5-forced runs are generally lower than those from CERRA-forced runs. The WFP has minimal impact on wind speed magnitude. When comparing CERRA-2d1kmMYnoFP with CERRA-2d1kmSH, the scale-aware SH scheme results in lower FLLJ wind speeds than the MYNN 2.5 scheme. Additionally, in comparing CERRA-2d1kmMYFP with CERRA-1d1kmMYFP, the single domain configuration produces consistently lower wind speeds than the nested domain configuration. Furthermore, the SCF is more distinct in the nested domain configuration compared to the single domain configuration.

To gain further insights, a transect line AB (shown in red) was drawn perpendicular to the frontal surface, equivalent to dissecting the frontal cross-section, which serves as a reference for wind speed cross-sectional analysis. The transect normal



**Figure 7.** Evolution of wind speed profiles from the five WRF simulations, for case 1 (left panel) and case 2 (right panel), at LOT2 location. The profiles are averaged over an hour during 03:00 to 04:00 (top panel), 04:00 to 05:00 (middle panel), and 05:00 to 06:00 (bottom panel).





wind speed is equivalent to front-parallel wind speed, shown in 8(middle panel) for case 1 and 9(middle panel) for case 2. The abscissa measures distances (km) along the transect line AB to/from the LOT2 location.

In both cases, regardless of the modeling parameters chosen, a clear and significant wind speed gradient is visible at the LOT2 location (0 km), indicating the presence of the SCF. Ahead of the SCF, from a distance of 10 km, there is a layer of wind speed maxima (jet core) in the vertical direction, indicative of an FLLJ. The choice of IC/BCs, PBL, and domain configuration appears to influence the FLLJ core strength and height.

In case 1, the ERA5 IC/BCs produced a weak core between 300-500 m altitude, while in case 2, no distinct core was formed. In contrast, the CERRA IC/BCs generated a clear jet core in both cases. For case 1, the core height is 400-500 m within 50 km of the SCF, with a core strength of 22-23 m/s. Beyond this range, orographic interactions cause the jet core to rise, reaching 25 m/s at about 100 km distance. In case 2, the core is at 400-600 m with a strength of 21 m/s, and similarly rises to higher altitudes, reaching 22 m/s at approximately 150 km due to orographic interactions.

Comparing CERRA-2d1kmMYnoFP and CERRA-2d1kmSH, the scale-aware SH scheme simulated a higher FLLJ core height in both cases. The core strength is lower in case 1 and higher in case 2 compared to MYNN 2.5 scheme simulations. Additionally, wind speeds below 200 m are consistently lower in the SH scheme.

For CERRA-2d1kmMYFP versus CERRA-1d1kmMYFP, the single-domain configuration resulted in a higher core height and higher wind speeds below 200 m, producing a more diffused jet nose profile across both cases.

From the cross-sectional analysis, it is clear that the front-parallel component mainly contributes to the occurrence of FLLJ, which is in line with the findings of Browning and Pardoe (1973). On the other hand, the front-parallel winds behind the frontal boundary maintain a magnitude of 8-12 m/s in case 1, contributing to the observed directional inclination seen in the wind barbs, whereas a magnitude of 0 m/s is seen in case 2, as evident from the wind barbs.

The transect-parallel wind speed is equivalent to front-normal wind speed, shown in 8(bottom panel) for case 1 and 9(bottom panel) for case 2. Interestingly, wind speeds ahead and behind the frontal surface are in the opposite direction and are exactly wedged by the frontal boundary. Findings of Browning and Harrold (1970) confirm that the parallel component behind the front aligns with the frontal surface's movement, while the component ahead of the front directs towards the frontal surface (which actually ascends to the upper levels, not displayed in these figures), a behavior aptly captured by all the simulations.

In both cases, the front-normal winds ahead of the boundary are much stronger than the winds behind the boundary. These have two diverse effects: one, winds ahead of the boundary contribute to strengthening FLLJ intensity, and two, winds behind the boundary result in a drastic wind speed drop, ultimately leading to a severe ramp event.

To put together the significance of front-parallel and normal circulations on wind power production, consider any event. An approaching cold front brings FLLJ along with it, spanning hundreds of kilometers ahead of the frontal surface and resulting in maximized wind power production for several hours. Within the vicinity of the frontal surface, the front-normal component also contributes to further intensifying the FLLJ. Once the front overpasses a wind farm, wind speed drops from an intense FLLJ magnitude to the magnitude of winds trailing the front, resulting in an intense ramp event.

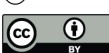

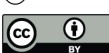

**Figure 8.** Illustration of horizontal wind speed from the five WRF runs: ERA5-3d1kmMYFP (1st row), CERRA-2d1kmMYFP (2nd row), CERRA-2d1kmMYnoFP (3rd row), CERRA-2d1kmSH (4th row), and CERRA-1d1kmMYFP (5th row), for case 1, during ramp time instances of the corresponding WRF run. Left panel: spatial distribution of wind barbs at 200 m level. The ramp instances of the corresponding WRF run are listed as centre title. A transect line AB in red color is drawn perpendicular to the frontal surace, while the buoy locations LOT1 and lOT2 are illustrated with black stars. Middle panel: Wind speed normal to the transect line AB, which gives an idea about the front parallel wind speed. Right panel: Wind speed parallel to the transect line AB, which gives an idea about the front normal wind speed. The abscissa in middle and bottom panels is the distance (km) along the transect line AB to/from LOT2 location, whereas the ordinate is vertical level (m).



**Figure 9.** Same as Figure 8, but for case 2, at the ramp instances of the corresponding simulation.





**Table 3.** Overview of the ramp in wind power in terms of intensity (%) and timing (in parenthesis), at 1 hr time-scale, obtained from the Belgium offshore wind farm inhourse day-ahead forecast and the GFS-3d1kmMYFP simulation, for cases 1 and 2.

| Model | Case 1 | Case 2 |
|---|---|---|
| **Grid-measured** | **54.3 (06:30)** | **88.2 (07:00)** |
| **Elia-forecast** | 33.2 (07:45) | 17.7 (10:00) |
| **GFS-3d1kmMYFP** | 73.1 (05:15) | 64.2 (07:45) |

## 4.5  Real-time forecasting

In this section, we investigate whether extreme ramp events associated with FLLJ could have been predicted in a real-time forecast scenario. The initial and boundary conditions provided by reanalysis datasets (e.g., ERA5 and CERRA) tend to be very accurate due to extensive data assimilation. However, in the context of real-time forecasting, such high-fidelity boundary conditions are not available and operational forecast data from global models (e.g., Global Forecasting System (GFS)) can be used. To assess this scenario, cases 1 and 2 are forecasted using a similar configuration as ERA5-3d1kmMYFP, except with

GFS real-time forecast data provided as initial and boundary conditions. The model forecasted wind power is compared with the grid-measured and Elia day-ahead forecasted power for cases 1 and 2, as shown in Figure 10. We would like to point out that the modeling details of Elia day-ahead forecast are not known to us, only the forecasted power values are available. Surprisingly, the WRF model was able to forecast the extreme ramp event in both cases, albeit with some degree of timing mismatch. In contrast, the Elia forecast shows a gradual ramp-down event in case 1, while no ramp was observed in case

2. Additionally, the model accurately forecasted the pre-ramp wind power maxima, implying that the FLLJs are being well forecasted in both cases.

The ramp statistics are computed at a 1-hour time scale and are presented in Table 3. In case 1, the model forecasted the ramp 1 hour and 15 minutes in advance with an overestimation in intensity, whereas the Elia forecast delayed the ramp by 1 hour and 15 minutes with an underestimation in intensity. In case 2, the model forecasted ramp was delayed by 45 minutes

with an underestimation in intensity, whereas no such ramp has been detected in the Elia forecasted power (a 17% change in 1 hour is not typically considered a ramp).

To compare, the ramp intensity from the model forecasts is of a similar magnitude to that of the CERRA-2d1kmMYFP simulation for case 1 and to that of the CERRA-1d1kmMYFP for case 2. This implies that extreme ramps can be forecasted well one day in advance, with an accuracy comparable to that of reanalysis dataset simulations.

## 4.6  Generalization across multiple cases


So far, the results have been focused on the analysis of two cases. To generalize the robustness the WRF model in simulating the extreme ramp events associated with FLLJ, three more cases have been simulated using the CERRA-1d1kmMYFP config-



**Figure 10.** A comparison of wind power time series from the GFS-MYFP simulation and the Belgium offshore wind farm production and their Elia day-ahead forecast, during 0100 UTC to 1200 UTC on 21st of February, 2016, for case 1 (top panel) and during 0200 UTC to 1300 UTC on 4th of March, 2016, for case 2 (bottom panel).





**Table 4.** Overview of the ramp in wind power in terms of intensity (%) and timing (in parenthesis), at 1hr time-scale, obtained from the wind power produced by the Belgium offshore wind farm and the CERRA-1d1kmMYFP configuration, for cases 3, 4, and 5.

| Model | Case 3 | Case 4 | Case 5 |
|---|---|---|---|
| **Grid-measured** | 67 | 52.9 | 52.7 |
| | (09:15) | (22:00) | (06:30) |
| **CERRA-1d1kmMYFP** | 54.2 | 50 | 58.4 |
| | (07:15) | (21:15) | (06:15) |

uration. We choose only one configuration, due to lack of computational power. The synoptic charts and the measured wind power time-series of these additional cases are illustrated in Figures A1 to A3, showcasing extreme ramp events. A comparison
between the wind farm-measured and model-simulated wind power time-series is illustrated in Figure 11 for the three cases.

Clearly, the WRF model was able to simulate the extreme ramps in wind power but exhibited minimal to considerable discrepancies in ramp timing. In all cases, the model-simulated wind power shows a consistent maximum of 712 MW (rated capacity) during the pre-ramp period, likely attributed to the peak wind speeds associated with the FLLJ, while the measured wind power also exhibits peaks but below the rated capacity.

The ramp statistics are computed at a 1-hour time scale and are presented in Table 4. From these statistics, it is evident that the WRF model accurately replicated the ramp intensity in all events, with power ramps surpassing 50% within just 1 hour, signifying the extremity of the power ramps. In terms of ramp timing, the simulated ramps are in advance by 2 hours, 45 minutes, and 15 minutes in cases 3, 4, and 5, respectively. Nonetheless, these findings corroborate the robustness of the WRF model in simulating the extreme ramp events associated with the FLLJ.

## 5   Conclusion and recommendations

In this study, we utilized the WRF model to simulate FLLJ and the associated extreme ramp events to analyze the variability in ramp timing, intensity, and jet core strength from a wind energy perspective. We identified five cases where wind power dropped by more than 50% within 1 hour at the Belgium offshore wind farm. Two of these cases were selected for in-depth analysis, while the remaining three were used for generalization. We examined the sensitivity of various model configurations, including
different initial and boundary condition datasets, the activation of Fitch wind farm parameterization, and PBL schemes, to understand their impact on the FLLJ and the associated extreme ramp characteristics.

For basic climatological characterizations of FLLJs and associated extreme ramp events, the ERA5 or CERRA reanalysis datasets are highly beneficial due to their extensive coverage, long-term availability, and high accuracy. However, these datasets do not account for the wake effects of wind farms. To incorporate such effects, using the WRF model or other mesoscale

**Figure 11.** A comparison of wind power time series from the CERRA-1d1kmMYFP configuration and the Belgium offshore wind farm production. Top panel: during 0100 UTC to 1200 UTC on 9th of February, 2016, for case 3. Middle panel: during 1800 UTC on 9th to 0600 UTC on 10th of January, 2017, for case 4. Bottom panel: during 0100 UTC to 1200 UTC on 30th of January, 2017, for case 5.





models with an appropriate wind farm parameterization is recommended. Additionally, mesoscale simulations offer advanced diagnostics (e.g., turbulent kinetic energy) and, due to their high spatial resolutions, can effectively resolve coastal effects.

In this study, we found that the chosen modeling parameters significantly influence ramp characteristics and FLLJ synoptic features. The CERRA IC/BCs provided a better representation of the atmospheric flow compared to the ERA5 reanalysis, resulting in more accurate ramp timing, intensity, and FLLJ synoptic features. Activating wind farm parameterization (WFP) significantly impacted wind power output, as WFP models the interaction of individual wind turbines with the atmospheric flow and generates wind wakes in the vicinity of the wind farm. Due to wake generation, wind speeds with WFP were marginally lower than without WFP.

The scale-aware SH scheme produced a stronger FLLJ core at a higher altitude with a more defined and pronounced jet nose compared to the MYNN 2.5 scheme. However, wind speeds below 200 m were lower with the SH scheme than with MYNN 2.5. Finally, the single-domain configuration was more effective in simulating wind power ramp timing and intensity, consistently showing a higher core height than the nested configuration. Additionally, this configuration resulted in higher wind speeds below 200 m, leading to a more diffused jet nose profile.

Analysis of FLLJ synoptic characteristics during ramp instances revealed that the FLLJ phenomenon is primarily driven by the transect normal wind component, aligned parallel to the frontal surface. The heightened wind ramp intensity results from the interplay between minimal transect normal wind speed behind the front and the intensified FLLJ ahead of the front.

We observed that wind power ramp statistics can be reliably simulated using a single domain configuration, as confirmed by three additional cases. If this approach is applicable to other FLLJ and associated extreme ramp events, it could enable accurate simulations with significantly reduced computational costs. In real-time forecasting scenarios, extreme ramp events could be predicted one day in advance with accuracy comparable to reanalysis simulations, enhancing operational efficiency.

Before concluding, it is important to note that the enhanced understanding and forecasting capabilities of Frontal Low-Level Jets (FLLJs) and associated wind power ramp events extend beyond the wind energy sector. Improved insights into FLLJs are also crucial for grid operators and energy market regulators, who need to ensure system stability and manage the economic implications of sudden fluctuations in power generation.

## Appendix A: Additionl case studies

This section presents wind power time-series measured by the Belgium offshore wind farms during the three additional cases and the corresponding synoptic charts.

## A1 Case 3

On 9th February 2016, the wind farm experienced a 67% drop in it's measured wind power within 1 hour, beginning from 0915 UTC, as shown in Figure A1(top panel). The farm is seen producing a maximum wind power of 600 MW, for more than 9 hours period before experiencing the ramp event, suggesting the occurrence of peak wind speed during this period. Coinciding, the synoptic weather maps at 0000 UTC on 9th and 10th of February (bottom left and right panels of Figure A1, respectively)



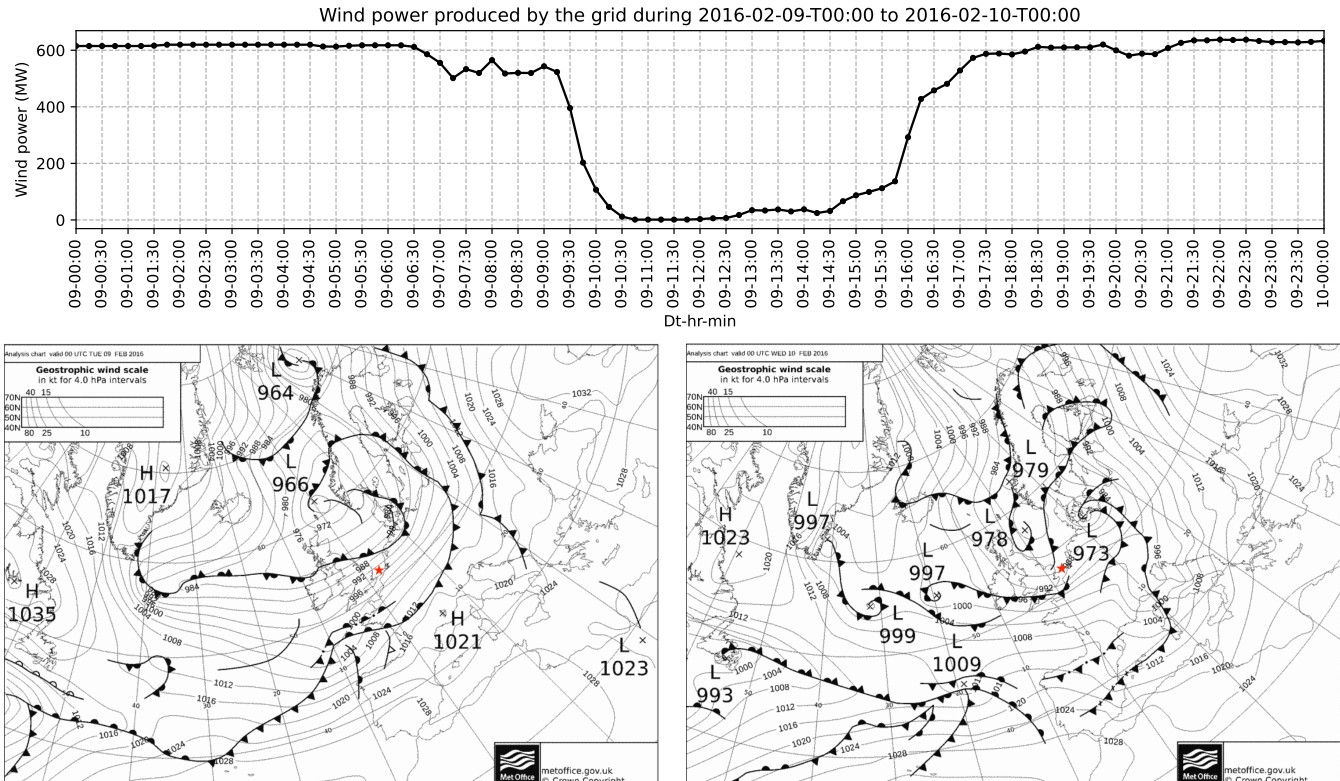

**Figure A1.** Same as Figure 1, but for case 3. The top panel shows the time series of wind power produced by the Belgian offshore wind farms, from 0000 UTC on the 9th to 1000 UTC on the 10th of February 2016, which depicts the extreme power ramp. Synoptic weather maps on 0000 UTC of 9th February 2016 (bottom left panel) and 0000 UTC of 10th February 2016 (bottom right panel) illustrating a cold front passing over the Belgium offshore wind farm (illustrated with red star).

clearly illustrating a cold front (dark line with triangles pointing the direction of frontal movement) overpassing the wind farm.

## A2 Case 4

495 On 9th January 2017, the wind farm experienced a 52% drop in it's measured wind power within 1 hour, beginning from 2200 UTC, as shown in Figure A2(top panel). The farm is seen producing a maximum wind power of 690 MW, for more than 4 hours period before experiencing the ramp event, suggesting the occurrence of peak wind speed during this period. Coinciding, the synoptic weather maps at 0000 UTC on 9th and 10th of January (bottom left and right panels of Figure A2, respectively) clearly illustrating a cold front (dark line with triangles pointing the direction of frontal movement) overpassing the wind farm.

500





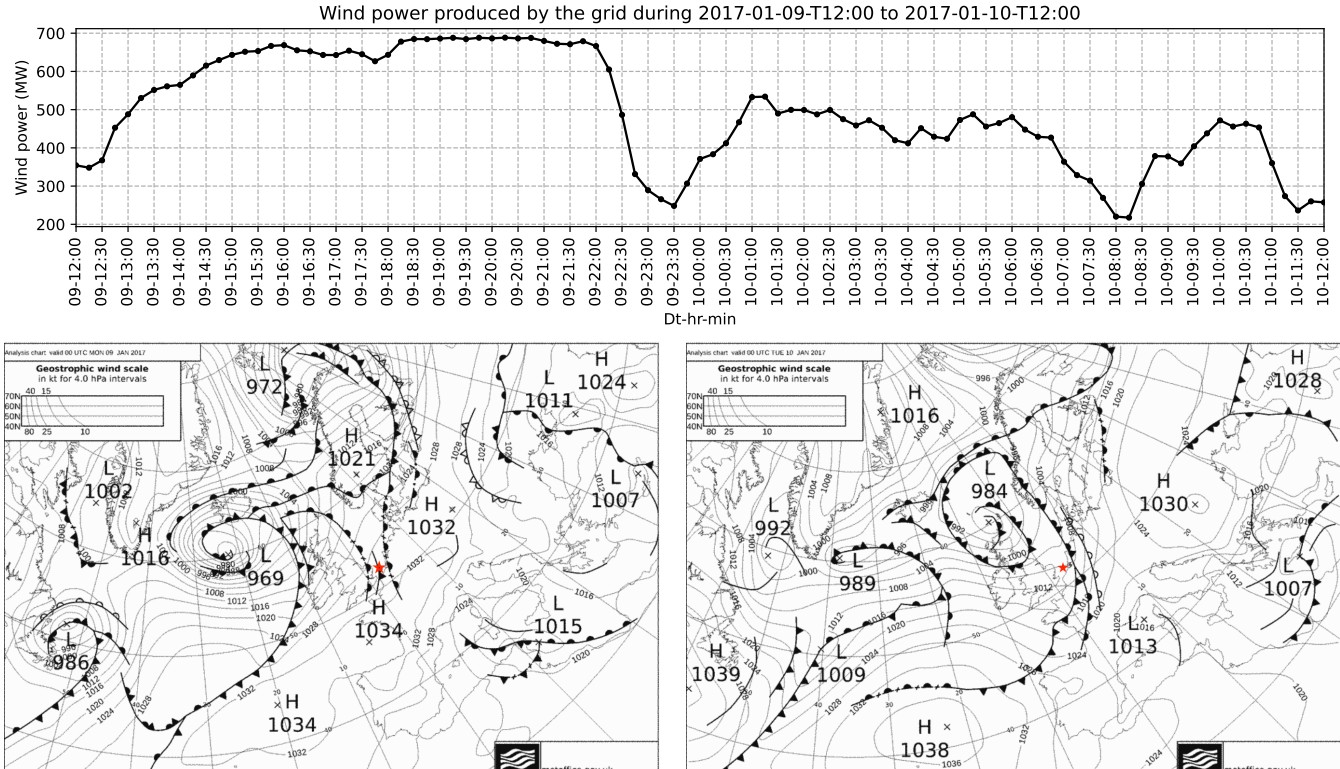

**Figure A2.** Same as Figure 1, but for case 4. The top panel shows the time series of wind power produced by the Belgian offshore wind farms, from 1200 UTC on the 9th to 1200 UTC on the 10th of January 2017, which depicts the extreme power ramp. Synoptic weather maps on 0000 UTC of 9th January 2017 (bottom left panel) and 0000 UTC of 10th January 2017 (bottom right panel) illustrating a cold front passing over the Belgium offshore wind farm (illustrated with red star).

## A3 Case 5

On 30th January 2017, the wind farm experienced a 52% drop in it's measured wind power within 1 hour, beginning from 0630 UTC, as shown in Figure A3(top panel). The farm is seen producing a maximum wind power of 670 MW, for more than 12 hours period before experiencing the ramp event, suggesting the occurrence of peak wind speed during this period. Coinciding, the synoptic weather maps at 0000 UTC on 30th and 31st of January (bottom left and right panels of Figure A3, respectively) clearly illustrating a cold front (dark line with triangles pointing the direction of frontal movement) overpassing the wind farm.

## Appendix B: Comparison between ERA5 and CERRA reanalysis

This section presents a comparison between ERA5 and CERRA datasets, for cases 1 and 2.

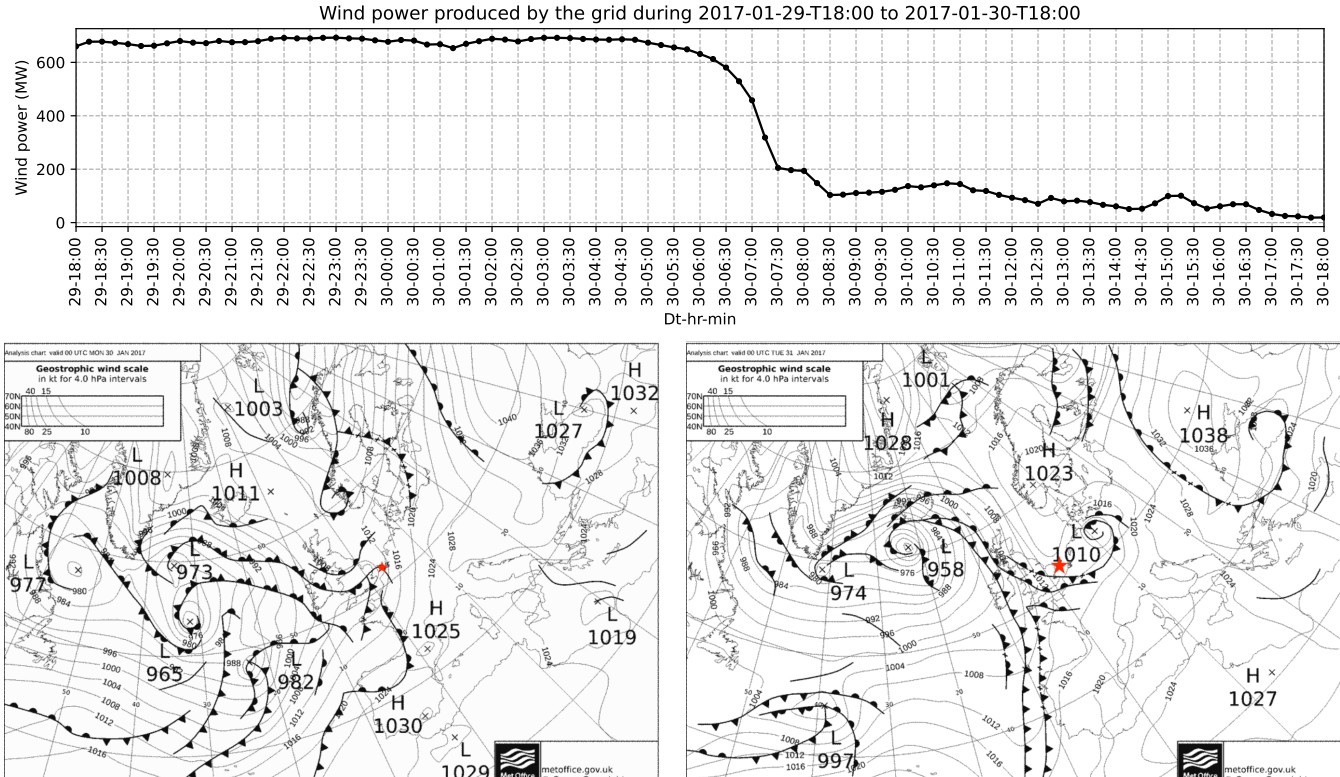

**Figure A3.** Same as Figure 1, but for case 5. The top panel shows the time series of wind power produced by the Belgian offshore wind farms, from 1800 UTC on the 29th to 1800 UTC on the 30th of January 2017, which depicts the extreme power ramp. Synoptic weather maps on 0000 UTC of 30th January 2017 (bottom left panel) and 0000 UTC of 31st January 2017 (bottom right panel) illustrating a cold front passing over the Belgium offshore wind farm (illustrated with red star).

## B1 Wind speed time-series

The ERA5 reanalysis provides 100 m level wind speed at hourly interval. In contrast, the CERRA reanalysis provides 100 m wind speed at three hourly analysis and hourly forecast. To mimic a continuous hourly dataset, we procured analysis at every three hours and the forecast at the next two hours. A comparison between these two wind speed datasets along with the lidar observations at LOT2 location are presented in Figure B1.

## B2 Wind profiles

The ERA5 and CERRA reanalysis datasets provide model level wind speed data, which could serve as wind speed profiles. However, we intend to visualize the wind speed profiles from the dataset we used as IC/BCs, which are indeed pressure level dataset. In doing so, we extracted the profiles from metfiles of ERA5 and CERRA runs' parent domain. The profiles are illustrated in Figure B2, for cases 1 and 2.





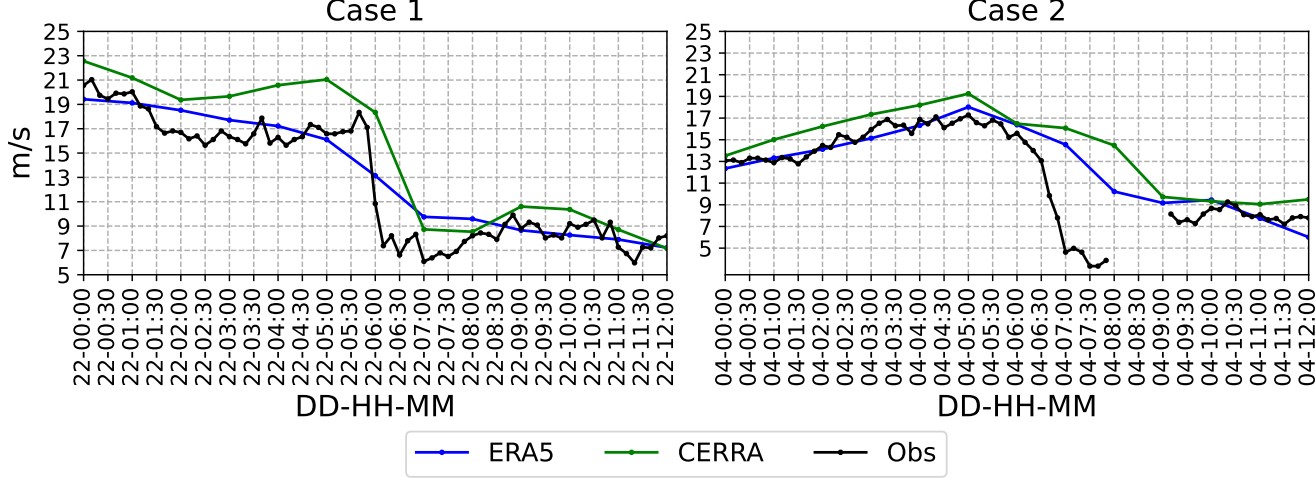

**Figure B1.** A comparison of 100 m wind speed time series from lidar observations, ERA5, and CERRA reanalysis. Left panel: during 0100 UTC to 1200 UTC on 21st of February, 2016, for case 1. Right panel: during 0200 UTC to 1300 UTC on 4th of March, 2016, for case 2.

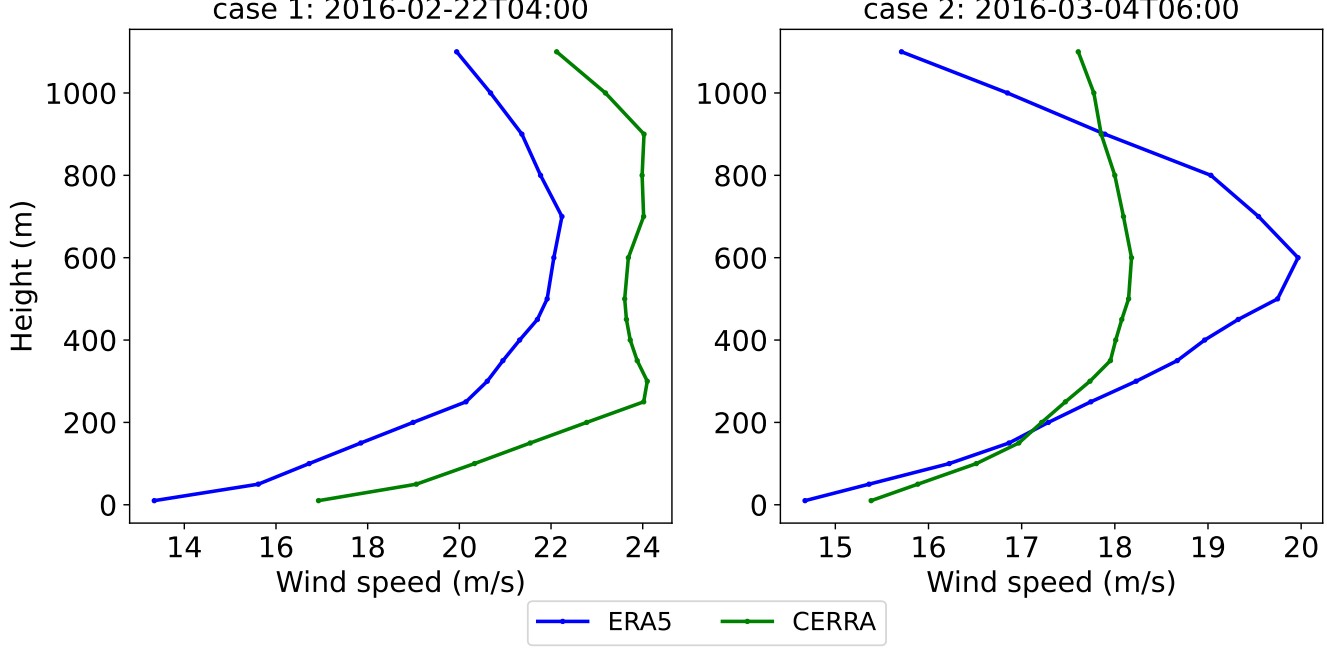

**Figure B2.** A comparison of wind speed profiles from the ERA5 and CERRA reanalysis, for case 1 (left panel) during 0500 UTC on 22nd Feb 2016 and case 2 (right panel) during 0600 UTC on 4th Mar 2016, at LOT2 location.





## Appendix C: Time-height cross-section of wind speed from the WRF runs

This section presents the time-height cross-section of wind speed from the five WRF runs along with the lidar observations at the LOT2 location, for cases 1 and 2.

### Appendix D: Wind wakes

This section presents a comparison of horizontal wind speed contours at 100 m level from the CERRA-2d1kmMYFP and CERRA-2d1kmMYnoFP simulations, at different time instances, for cases 1 and 2.

*Data availability.* The ERA5 and CERRA reanalysis are downloaded from ECMWF CDO, available at https://cds.climate.copernicus.eu/cdsapp#!/search?type=dataset.

*Author contributions.* HB is responsible for data analysis, model simulations, and manuscript writing. SB and GL contributed to manuscript organization, text revision, and result discussions.

*Competing interests.* Some authors are members of the editorial board of journal Wind Energy Science.

*Acknowledgements.* The authors acknowledge the use of computational resources of DelftBlue supercomputer, provided by Delft High Performance Computing Centre (https://www.tudelft.nl/dhpc). This EU-SCORES project has received funding from the European Union's Horizon 2020 research and innovation programme under grant agreement No 101036457. We extend our gratitude to Elia for publicly sharing the Belgian offshore wind power production data. We also thank TNO for providing the LiDAR observed data.



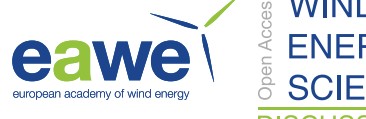

**Figure C1.** Time-height cross-sections of wind speed at the LOT2 location for cases 1 and 2. Top panel shows wind speed from lidar observations. Panels from 2nd row to the last show wind speed from the WRF simulations, ERA5-3d1kmMYFP, CERRA-2d1kmMYFP, CERRA-2d1kmMYnoFP, CERRA-2d1kmSH, and CERRA-1d1kmMYFP. A solid black line spanning through the time series indicates the presence of low level jet.





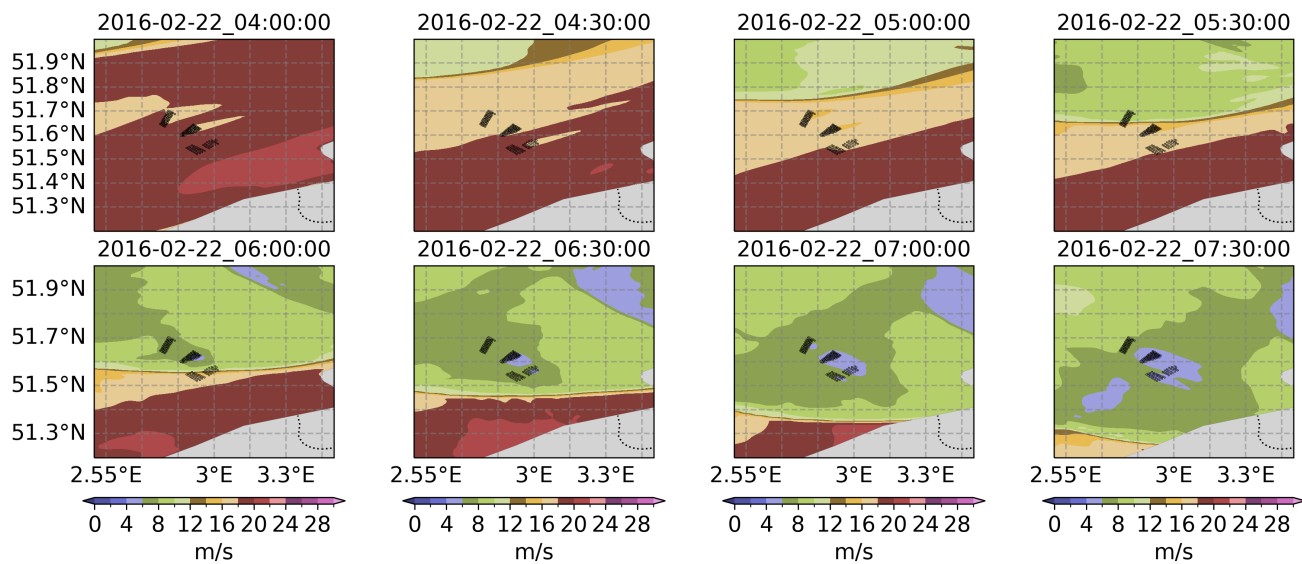

**Figure D1.** Illustrations of 100 m level wind speed from the CERRA-2d1kmMYFP simulation for case 1, at the instance of every 30 minutes during 0400 UTC to 0730 UTC on 22nd February, 2016.

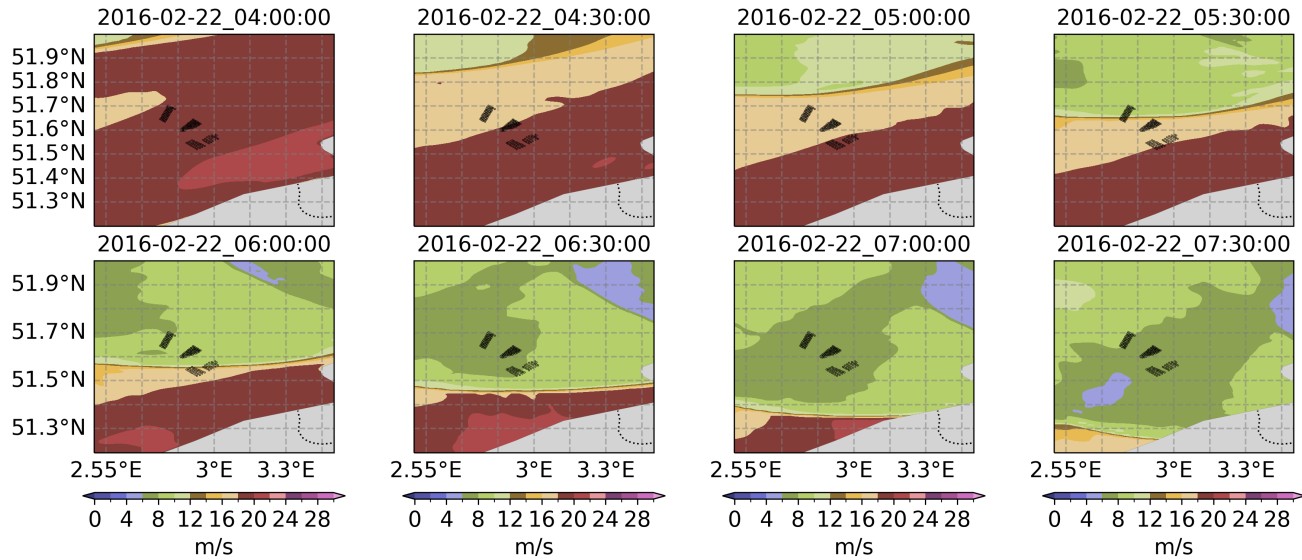

**Figure D2.** Same as Figure D1, but from the 2d1kmMYnoFP simulation.



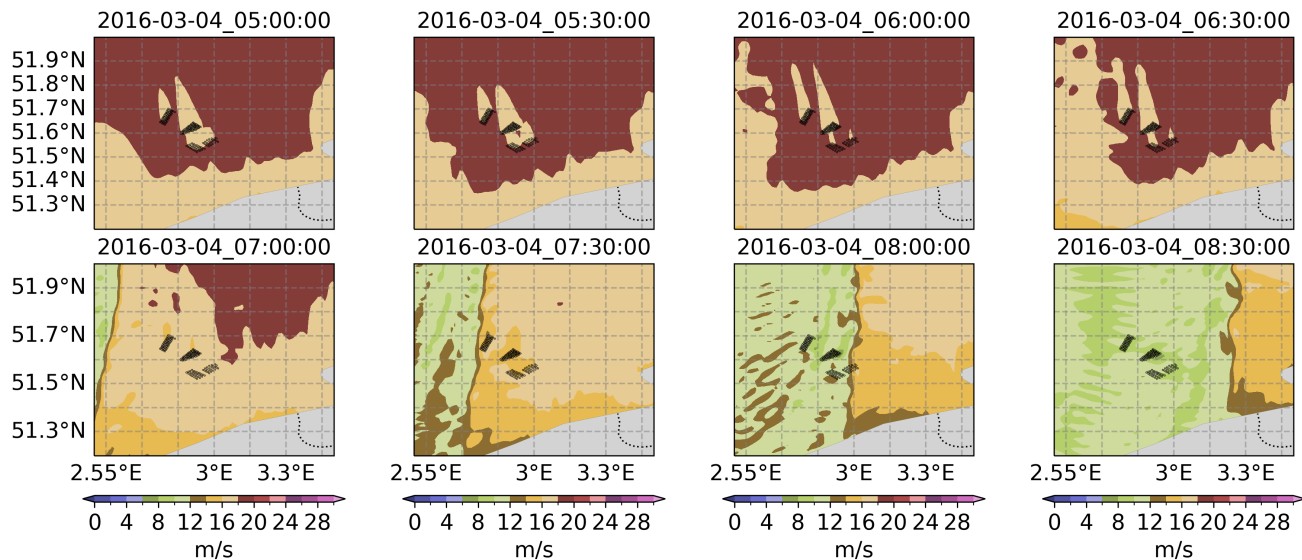

**Figure D3.** Illustrations of 100 m level wind speed from the CERRA-2d1kmMYFP simulation for case 2, at the instance of every 30 minutes during 0500 UTC to 0830 UTC on 4th March, 2016.

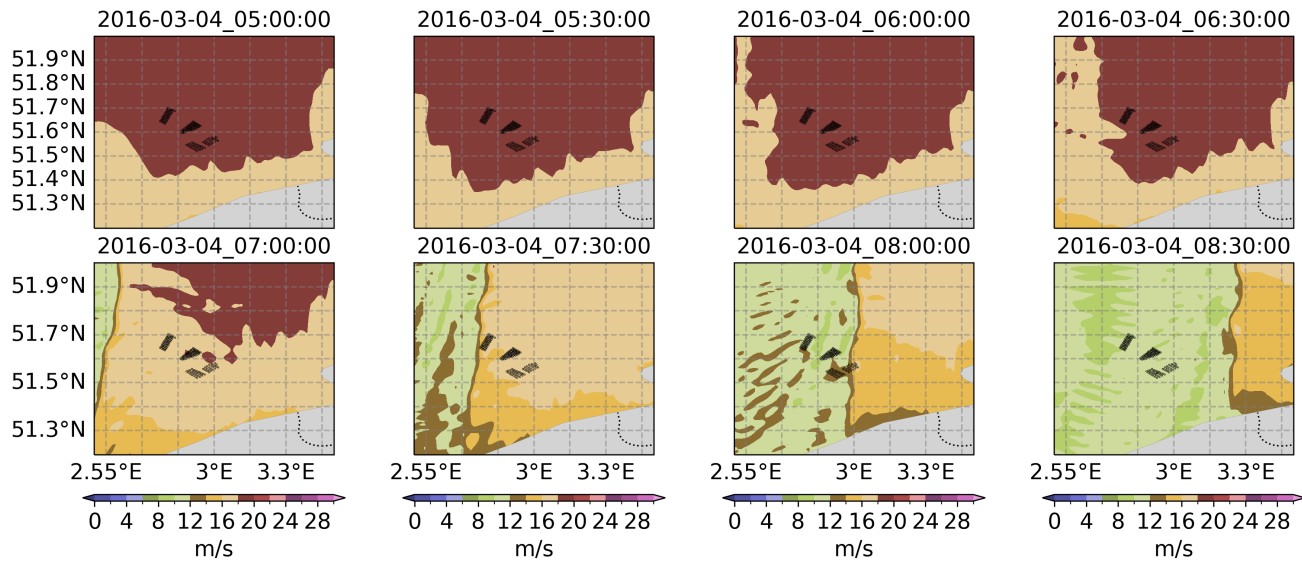

**Figure D4.** Same as Figure D3, but from the 2d1kmMYnoFP simulation.





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
