# Peer review of "Modelling Frontal Low-Level Jets and Associated Extreme Wind Power Ramps over the North Sea"

_Wind Energy Science, 2024_

## Author Comment (AC1)

Dear Reviewer 1, Reviewer 2, and the Associate Editor,

Thank you very much for constructive feedback on our manuscript. Your insightful comments have significantly contributed to improving the overall quality of the paper.

In response to the reviewers' comments, we have undertaken a substantial revision of the manuscript, incorporating several new simulations to strengthen the study. Specifically, we conducted additional sensitivity simulations to evaluate the influence of vertical resolution and spin-up time on model performance. Furthermore, we extended the WRF simulations using the CERRA-based configuration across three additional extreme ramp cases. These efforts have not only addressed the reviewers' specific concerns but have also enhanced the comprehensiveness of the manuscript.

Below, we provide detailed responses to each of the reviewers' comments, outlining the specific revisions and clarifications made in the manuscript.

**Reviewer 1**

**Formatting**

1. Figures 1 and 2 (and Figures A1, A2, and A3): add subplot letters (a), (b), and (c)

   Based on the suggestion from both the reviewers, all the figures have been updated with the appropriate subplot letters added, in the revised manuscript.

2. The date and time should follow the format: 25 July 2007 (dd month yyyy), 15:17:02 (hh:mm:ss).

   In the revised manuscript, figures 1, 2, A1, A2, and A3, are updated with time format being changed to #day #month #year #hour:#minute.

3. …The abbreviation "Fig." should be used when it appears in running text and should be followed by a number unless it comes at the beginning of a sentence, e.g.: "The results are depicted in Fig. 5. Figure 9 reveals that...".

   Throughout the revised manuscript, Figure changed to Fig., followed by the figure number.

4. Regarding the notation, if units of physical quantities are in the denominator, contain numbers, and are abbreviated, they must be formatted with negative exponents (e.g. 10 km h-1 instead of 10 km/h).

   In the revised manuscript, notation of units with quantities in the denominator has been formatted to negative exponent, such as m/s to m $s^{-1}$ and km/h to km $h^{-1}$.

**Specific comments and questions**

1. P4L119-120: It's not clear to me what "it's" refers to here

    In the sentence "The jet's acceleration is driven by the isallobaric term, with Coriolis torque and advective tendency terms contributing to its propagation perpendicular to the FLLJ.", it refers to the FLLJ itself. However, the additional terms "perpendicular to the FLLJ" mislead the reader. It should be "parallel to the frontal surface".

    The entire sentence has been rephrased as below for clarity, and is modified in the revised, at P4L119-120.

    The jet's acceleration is driven by the isallobaric term, while the Coriolis torque and advective tendency terms influence its propagation along the frontal surface and adjust its alignment relative to the frontal boundary.

2. P5L145: Why did you choose the two cases from the five as you did? What criteria did you use?

    The choice of cases is arbitrary. We selected two cases for a complete analysis while the rest of three cases for selected analysis.

3. P6L162: I assume the "forecast" experiment is using GFS? Perhaps it's obvious, but consider stating it explicitly here already

    In the revised manuscript, the forecast experiment is explicitly mentioned in the methods section as presented below.

    At P8L161-163:

    For the sensitivity analysis, we chose to vary the initial and boundary conditions, using ERA5 and CERRA datasets to represent hindcast experiments and GFS (Global Forecasting System) to represent the forecast experiment.

    At P10L195-201:

    The initial and boundary conditions provided by reanalysis datasets (e.g., ERA5 and CERRA) tend to be very accurate due to extensive data assimilation. However, in the context of real-time forecasting, such high-fidelity boundary conditions are not available, thus operational forecast data from global models (e.g., GFS) can be used. In the forecast experiment GFS-3d1kmMYFP, cases 1 and 2 are forecasted using a similar configuration as ERA5-3d1kmMYFP, except with GFS real-time forecast data provided as initial and boundary conditions. The GFS IC/BCs during the simulation

period are available at a three-hourly resolution and a horizontal grid spacing of about 30km.

At P11L243-244:

The GFS IC/BCs initialized at 1200 UTC on 21st February 2016 for case 1 and at 1200 UTC on 3rd March 2016 for case 2 are obtained for the forecast experiment.

4.  MYNN2.5: what "bl_mynn_*" settings were used? The defaults of WRF v4.4? MYNN2.5 can be quite different depending on this.

In our simulations we used the following settings for bl_mynn.

bl_mynn_tkebudget = 1,

bl_mynn_tkeadvect = .true.,

On the other hand, the other settings are unaltered, thus taken as their default settings.

5.  In a study focussing on capturing the timing of an event, it's surprising to me that you don't consider the influence of data assimilation (except to say that is one reason for the accuracy of re-analysis datasets). Why did you not test data assimilation and/or discuss this in the paper?

It is true that data assimilation can be a crucial factor, especially when it comes to real-time forecasting, where accurately initializing the model states is key to improving short-term predictions.

However, in this study, our main objective was to examine the sensitivity of modeling parameters and assess the ability of the WRF model to simulate frontal low-level jets and associated extreme ramp events. By focusing on the modeling aspects, we aim to better understand the dynamics and physical processes involved in the event simulation, separate from the impacts of assimilation techniques.

6.  You use 51 levels, why not more? Are you sure it's not sensitive to this?

In the revised manuscript, we have examined the influence of vertical levels on the simulations by selecting 101 vertical levels, with approximately 27 vertical levels within the first 1 km of height. Using this configuration and the CERRA-1km1dMYFP simulation strategy, the first two cases were simulated.

From the analysis, it was found that the vertical levels exert noticeable influence on the ramp timing and post-ramp wind speed. However, the performance of the 51 vertical levels configuration was better than that of 101 vertical levels, further justifying our choice of

vertical levels. The analysis is presented in Appendix B in the revised manuscript and is presented below for reference.

We have examined the influence of vertical levels on the simulations by selecting 100 vertical levels, with approximately 27 vertical levels within the first 1 km of height. Using this vertical resolution and the CERRA-1km1dMYFP configuration, the first two cases were simulated. In our study, the choice of 50 vertical levels was adopted from Nunalee and Basu (2014), where in the case of coastal LLJs, the authors reported reduced jet strength and lower jet core height with increased vertical levels, deviated from the observations. Similar to their findings, we noticed the vertical levels exert considerable influence on the ramp timing and marginal influence on the ramp intensity, as shown in Fig. B1. However, we also recognize the limitations of our findings, since they are based on two simulations focused specifically on FLLJ cases.

[Figure]

Figure 1: A comparison of wind power time series from the WRF model simulations of CERRA-1d1kmMYFP configuration, with 51 vertical levels and 101 vertical levels, and the Belgium offshore wind farm production (a) During 0100 UTC to 1200 UTC on 21st of February, 2016, for case 1. (b) During 0200 UTC to 1300 UTC on 4th of March, 2016, for case 2.

7. You used 6 hr of spin-up. Are you sure the model has enough time to develop these strong weather events? Perhaps the poor performance of some of the experiments is due to insufficient spin-up time. See e.g. Lui et al. (2023)

To evaluate the sensitivity with respect to spin-up time, we conducted simulations for the two cases using the CERRA-1d1kmMYFP configuration, one with a 6-hour spin-up and another with a 12-hour spin-up. The results revealed that the wind power output from both simulations was identical, indicating that a 6-hour spin-up was sufficient for the development of the FLLJ cases. This confirms the adequacy of the chosen spin-up duration for accurately capturing the dynamics of these events. The following analysis is presented in Appendix C in the revised manuscript.

> To evaluate the sensitivity of model accuracy to the choice of spin-up time, we conducted simulations for two cases using the CERRA-1d1kmMYFP configuration: one with a 6-hour spin-up (original simulation duration) and another with a 12-hour spin-up (starting 6 hours prior to the original duration). Figure C1 presents wind power time-series from both simulations, compared with the measured power output for Cases 1 and 2. The results show that the wind power output from both simulations is identical, indicating that a 6-hour spin-up is sufficient for the development of the FLLJ cases. However, it is highly unlikely for simulations initialized with different IC/BCs to produce identical results. To further verify this, we compared the time-height cross-section of wind speed from the two simulations, along with lidar observations at the LOT2 location, for Cases 1 and 2, as shown in Fig. C2. The comparison reveals that the simulations are nearly identical, with no discernible differences in wind speed, except for minor variations marked by sky-blue circles in both cases. Since no differences are observed in wind speed below the 100 m level, the wind power outputs from the two simulations also remain identical, as shown in Fig. C1. These findings confirm that the chosen 6-hour spin-up duration is adequate for accurately capturing the dynamics of these events.

[Figure]

*Figure 2: A comparison of wind power time series from the WRF model simulations of CERRA-1d1kmMYFP configuration, one with 6 hr spin-up and another with 12 hr spin-up, and the Belgium offshore wind farm production. Top panel: during 0100 UTC to 1200 UTC on 21st of February, 2016, for case 1. Bottom panel: during 0200 UTC to 1300 UTC on 4th of March, 2016.*

[Figure]

*Figure 3: Time-height cross-section of wind speed difference between the CERRA-1d1kmMYFP simulations with 6hr and 12hr spin-up, at the LOT2 location for cases 1 and 2.*

8.  Figures 4-10: Why not show the results from the datasets used for forcing data: ERA5, CERRA, and GFS? It would be more convincing to show that downscaling is needed if I could see the reference data as well.

We appreciate the reviewer's suggestion to include the results from the forcing datasets (ERA5, CERRA, and GFS) in Figures 4–10. However, we are unable to include these results in the manuscript for the following reasons.

We did analyze wind speeds from ERA5 and CERRA at 100 m and from GFS at the 975 hPa pressure level, comparing them with lidar observations, which are shown in Fig. D1, and are presented here for reference. Our analysis revealed that while GFS follows the overall trend of observed wind speeds, it exhibits a clear overestimation in magnitude. Though the GFS data at the current time exist at an hourly temporal resolution, it is only available at a 3-hourly resolution for 2016-17 years. Due to the coarse temporal resolution, it is challenging to quantify ramp statistics, particularly as extreme ramps associated with frontal low-level jets occur on scales of minutes. This further demonstrates that dynamical downscaling is indispensable for understanding extreme wind ramps at sub-hourly scales, which are critical for wind power applications.

Furthermore, and most importantly, CERRA and ERA5 provide wind speeds at 100 m above ground level, while GFS provides wind speeds at the 975 hPa pressure level. However, four of the five wind turbine types operating in the wind farm have hub heights below 100 m, making it challenging to accurately approximate wind power production using the forcing datasets.

Additionally, most of the analysis from WRF simulations in this study was conducted at temporal scales of 10–15 minutes, whereas the temporal resolutions of the forcing datasets range from 1 to 3 hours. This mismatch in temporal resolution further limits the utility of the forcing data in capturing rapid changes and short-term wind fluctuations associated with extreme ramps.

The ERA5 reanalysis exists at a spatial resolution of 0.25°, the CERRA analysis exists at a spatial resolution of 5.5 km, and the GFS forecast exists at a spatial resolution of 0.25°. Due to these coarse spatial resolutions, the entire wind farm fits within a single grid cell for ERA5 and GFS, while only a few grid cells from CERRA cover the wind farm. These spatial limitations hinder the representation of the wind power output generated by 182 turbines within the wind farm. This is critical, as the heterogeneity in turbine locations and wind conditions cannot be resolved with the coarser spatial scales of the forcing data.

Including the analysis from the forcing datasets would also increase the number of outputs by three, significantly complicating the visual presentation of Figures 4–10. We believe excluding the forcing data maintains the clarity and focus of the manuscript while strengthening the argument for the necessity of dynamical downscaling to capture sub-hourly wind variability.

The justification regarding the necessity of dynamical downscaling is incorporated at PL in the revised manuscript.

[Figure]

Figure 4: A comparison of 100 m wind speed time series from lidar observations, ERA5, and CERRA reanalysis. The GFS wind speed time series is extracted at 975 hPa level. Left panel: during 0100 UTC to 1200 UTC on 21st of February, 2016, for case 1. Right panel: during 0200 UTC to 1300 UTC on 4th of March, 2016, for case 2.

9.  The range of values in the colormap makes some of the plots difficult to it difficult to interpret. For example, do you really need to include values up to 30 m/s in figures D1-4. In Fig. C1, the breaks in the colormap seem to be inconsistent (sometimes 2 m/s, sometimes 1 m/s)

In the revised manuscript, the color levels of Figures F1-4 are set according to the minimum and maximum of the wind speed, at an interval of 1 m/s.

For Fig. E1, the color levels are set from 4 to 14 m/s at an interval of 2 m/s and from 15 to 28 m/s at an interval of 1 m/s. This gives clarity in visualizing the wind speed contours for cases 1 and 2.

10. Do you use the ERA5 pressure-levels or model-levels?

We have used the ERA5 pressure level data as initial and boundary conditions.

11. In P14L298-300: What exactly are you arguing here? that WRF cannot generate sharp gradients from coarse ERA5 boundary-condition data? You say CERRA constitutes better BCs, but it is itself based on ERA5.

In the referenced passage, we argue that while CERRA is derived from ERA5, it incorporates additional data assimilation and a significantly higher horizontal resolution (5.5 km vs. 31 km), which enhances its representation of mesoscale features. This improved resolution and assimilation process allow CERRA to better capture sharp gradients, such as those associated with the Frontal Low-Level Jet, providing more accurate boundary conditions for WRF simulations.

12. P23L423: Why are you surprised that the model captured the event? Please elaborate on why it's surprising

The intention was not to express surprise at the WRF model's ability to forecast the ramp and extremity of the FLLJ but rather to highlight the contrasting performance of the Elia forecast. Specifically, while the WRF model captured the ramp events in both cases, albeit with some timing mismatch, the Elia forecast either predicted a gradual ramp-down (case 1) or did not capture the ramp at all (case 2). This discrepancy was noteworthy because it suggests potential challenges in operational forecasting of such extreme events. We have revised the text to clarify this point and to better reflect the intended message without any exaggeration. The revised manuscript is updated with the following text, at P22L443-444.

> We note that the WRF model was able to forecast the extreme ramp event in both cases, though with some degree of timing mismatch.

13. Figure 11: Unfortunately, only the CERRA-1d1kmMYFP was included here, it would have been more convincing to show that the trend from cases 1 and 2 continues here

We appreciate your interest in extending the comparison to demonstrate the trends from Cases 1 and 2 within this figure. The three additional cases have been simulated using the CERRA-based configurations, namely CERRA-2d1kmMYFP, CERRA-2d1kmMYnoFP, CERRA-2d1kmSH, and CERRA-1d1kmMYFP. The following analysis, figure, and table are incorporated in the revised manuscript, at P26L463-476.

> The results demonstrate that the WRF model successfully simulates the extreme wind power ramps associated with FLLJs, though the specific modeling configurations significantly influence the wind power time series. All simulations capture the strong

pre-ramp wind power at the 720 MW rated capacity, likely due to peak wind speeds during the FLLJ, as well as the post-ramp wind power output, which aligns with observed values. The observed strong drop in wind power, representing the extreme ramp, is consistently simulated across cases; however, discrepancies in ramp intensity and timing persist, varying across configurations. The ramp statistics from the four CERRA-based configurations for three cases are computed at a 1-hour time scale and are presented in Table 4. The CERRA-1d1kmMYFP simulation shows superior performance, with the power ramps surpassing 50% within just 1 hour, signifying the extremity of the power ramps. In terms of ramp timing, the simulated ramps are in advance by 2 hours, 45 minutes, and 15 minutes in cases 3, 4, and 5, respectively. This indicates the robustness of CERRA-1d1kmMYFP in accurately capturing ramp timing and intensity for cases 4 and 5. In case 3, while the timing is simulated with a 2-hour lead, the ramp intensity is closely represented compared to other configurations. Other simulations exhibit larger temporal shifts in cases 4 and 5, and underpredicted intensities in cases 3 and 4. Nonetheless, these findings corroborate the robustness of the WRF model in simulating the extreme ramp events associated with the FLLJ, and the better predictability of the CERRA-1d1kmMYFP modeling configuration.

[Figure]

Figure 5: A comparison of wind power time series from the CERRA-2d1kmMYFP, CERRA-2d1kmMYnoFP, CERRA-2d1kmSH, and CERRA-1d1kmMYFP configurations and the Belgium offshore wind farm production. (a): during 0100 UTC to 1200 UTC on 9th of February, 2016, for case 3. (b): during 1800 UTC on 9th to 0600 UTC on 10th of January, 2017, for case 4. (d): during 0100 UTC to 1200 UTC on 30th of January, 2017, for case 5.

Table 1: Overview of the ramp in wind power in terms of intensity (%) and timing (in parenthesis), at a 1-hour time-scale, obtained from the wind power produced by the Belgium offshore wind farm and various CERRA configurations, for cases 3, 4, and 5.

| Model | Case 3 | Case 4 | Case 5 |
|---|---|---|---|
| Grid-measured | 67.0 (09:15) | 52.9 (22:00) | 52.7 (06:30) |
| CERRA-2d1kmMYFP | 35.2 (06:00) | 53.3 (20:45) | 55.8 (05:15) |
| CERRA-2d1kmMYnoFP | 47.1 (08:30) | 33.4 (20:45) | 57.6 (05:30) |
| CERRA-2d1kmSH | 52.9 (08:30) | 29.3 (20:30) | 51.7 (06:00) |
| CERRA-1d1kmMYFP | 54.2 (07:15) | 49.9 (21:15) | 58.4 (06:15) |

14. P27L463: Same as above, you say CERRA provides better BCs than ERA5 to capture timing and intensity, but CERRA is based on downscaling from ERA5, so perhaps the problem is not ERA5 BCs but the downscaling and e.g. lack of data assimilation?

In continuation, as stated in response 5, while CERRA is based on downscaling from ERA5, it provides better boundary conditions (BCs) due to its higher resolution (5.5 km) and the frequent incorporation of observational data through data assimilation techniques on a regional scale. These factors enhance CERRA's ability to represent mesoscale processes, allowing for more accurate reproduction of the timing and intensity of atmospheric phenomena like frontal low-level jets.

That said, we acknowledge that data assimilation could significantly impact the results, particularly for real-time simulations and improving the initial state of the atmosphere. However, data assimilation is out of scope for the current study, since many different approaches exist, such as 3DVAR, 4DVAR, Kalman filter, which perform differently as per the literature. In this study, our primary focus was on evaluating the model's sensitivity towards different modeling configurations and examining the differences between using higher and lower resolution IC/BCs.

**Technical corrections**

1. See "formatting" part

   We have updated the revised manuscript with appropriate formatting in accordance with the journal standards.

2. P13L275: time-serires -> time-series

   The typo has been corrected in the revised manuscript.

**Reviewer 2**

**Major comments**

1. It is unclear how the timing of a down-ramp is determined. While Equation 1 defines the intensity of the ramp across different scales, what thresholds are used to identify the ramp? Additionally, is any preprocessing (e.g., smoothing) necessary to avoid false alarms?

   The timing of a ramp in our study was determined using Equation 1, where the intensity of a ramp is calculated as the difference between the wind farm's power output at time $t + \Delta t$ and $t$, normalized by the wind farm's capacity. If this normalized difference exceeds a threshold, it is categorized as a ramp at the time instance t. For identifying ramp cases, we used a $\Delta t$ of 60 minutes (1 hour), adhering to a 25% drop in power output within this timeframe. This threshold ensured we focused on significant ramps associated with frontal cold fronts.

   Once the ramp cases were identified, further analysis was performed on shorter timescales of 15 minutes ($\Delta t = 15$) and 30 minutes ($\Delta t = 30$), using these time intervals to examine the intensity and dynamics of power drops within these cases. However, it's important to note that the primary criterion for identifying the ramps was the 25% drop in 1 hour, and no specific thresholds were applied for the 15-minute and 30-minute timeframes.

   Additionally, we did not employ any preprocessing or smoothing to avoid false alarms. The application of the 25% threshold for the 1-hour timeframe provided a robust basis for identifying extreme ramp events linked to frontal passages and associated wind dynamics.

2. The introduction highlights that unforecasted wind power down-ramps can lead to significant profit losses for farm owners. What is the critical timeframe within which an effective plan can be made in advance? Does a one-day forecast fall within this time window?

   Unforecasted wind power down-ramps can indeed lead to significant profit losses for wind farm operators, primarily due to challenges in maintaining grid stability and aligning power supply with demand. The critical timeframe for effective planning largely depends on the operational and grid requirements. For most grid operations, lead times of several hours to one day are considered crucial to implement effective mitigation strategies, such as ramping up alternative power sources, adjusting grid reserves, or curtailing demand.

In our study, we focused on extreme down-ramps associated with frontal cold fronts, which often develop and evolve over mesoscale temporal and spatial scales. These ramps can sometimes be anticipated within a 12 to 24-hour forecast window, depending on the accuracy of the mesoscale weather prediction models. Therefore, a one-day forecast does fall within the actionable timeframe, especially for operational wind energy forecasting systems. However, shorter lead times (e.g., 1–6 hours) are typically more critical for real-time adjustments and grid stability, where higher-resolution forecasting tools become essential.

3. When comparing the runs with and without WFS, does the way calculating wind power influence the results? For instance, the choice of the power curve.

The wind power calculation within the WRF model is governed by the turbine's power and thrust curves, which are provided by the manufacturer. For reference, the thrust and power curves of the wind turbines operating in the wind farm are provided below, referenced from the study of Li and Basu (2021). Within the WRF model with wind farm parameterization (WFP) activated, wind speed decreases downstream due to the conversion of kinetic energy into electric energy, by the turbines, thus altering the wind speed within the vicinity of the wind farm. The change in wind speed is governed by both power and thrust curves, one correlates wind speed with power, and other correlates momentum deficit with wind speed. On the other hand, offline wind power calculation for simulations without WFP activated only utilize the power curves, thus power output will be overestimated compared to with WFP.

Nonetheless, from the power curves, it is evident that different turbines output different wind power for a certain wind speed, thus, the choice of power curve influences the wind power calculations, with and without WFP.

**TABLE C1**    Sources of power and thrust curves

| Turbine | Power Curve | Thrust Curve |
| --- | --- | --- |
| Vestas V90 3 MW | Source: manufacturer | Source: Bot, [57] |
| Vestas V112 3 MW | Source: manufacturer | Source: manufacturer |
| Senvion (formerly REPower) 5 MW | Source: windPRO | Source: windPRO |
| Senvion 6.2 MW | Source: manufacturer | Proxy: Senvion 5 MW |
| Alstom 6 MW | Proxy: ECN 6 MW[58] | Proxy: ECN 6 MW[58] |

[Figure]

**FIGURE C1**    Power curves (left panel) and thrust curves (right panel) of the turbines from the Belgian offshore wind farms [Colour figure can be viewed at wileyonlinelibrary.com]

Li, B., Basu, S., Watson, S. J., & Russchenberg, H. W. (2021). Mesoscale modeling of a "Dunkelflaute" event. *Wind Energy*, *24*(1), 5-23.

4. I am not fully convinced that CERRA-1d1kmMYFP outperforms CERRA-2d1kmMYFP or other CERRA-forced runs in simulating wind speed and direction in Figure 5 and Figure 6. CERRA-1d1kmMYFP produces a much more gradual change in wind speed and direction. The second pick at around 09:30 in case one is not captured in CERRA-1d1kmMYFP. As for the timing of wind power down ramp, a larger offset is also observed in other cases such as case 3.

We agree with the reviewer that the CERRA-1d1kmMYFP simulation of wind speed and direction during the extreme ramp in case 1 is not accurate. Specifically, the simulated wind speed and direction changes are more gradual, and the secondary peak around 09:30 in case 1 is not well captured. Please note that, the inaccuracies in wind speed in the "rated" portion of the power curve do not influence wind power output, which is evidenced in CERRA-1d1kmMYFP configuration exhibiting high accuracy in simulating the wind power output, closely matching the measured values, including the ramp-down intensity and timing in cases 1 and 2.

To further quantify the robustness of CERRA-1d1kmMYFP in simulating wind power during extreme ramp-down events associated with FLLJs, we conducted additional simulations for three more cases (cases 3, 4, and 5) using the model configurations: CERRA-

2d1kmMYFP, CERRA-2d1kmMYnoFP, CERRA-2d1kmSH, and CERRA-1d1kmMYFP. The analysis and the newly incorporated text in the revised manuscript at P26L463-476, are provided below.

The results demonstrate that the WRF model successfully simulates the extreme wind power ramps associated with FLLJs, though the specific modeling configurations significantly influence the wind power time series. All simulations capture the strong pre-ramp wind power at the 720 MW rated capacity, likely due to peak wind speeds during the FLLJ, as well as the post-ramp wind power output, which aligns with observed values. The observed strong drop in wind power, representing the extreme ramp, is consistently simulated across cases; however, discrepancies in ramp intensity and timing persist, varying across configurations. The ramp statistics from the four CERRA-based configurations for three cases are computed at a 1-hour time scale and are presented in Table 4. The CERRA-1d1kmMYFP simulation shows superior performance, with the power ramps surpassing 50% within just 1 hour, signifying the extremity of the power ramps. In terms of ramp timing, the simulated ramps are in advance by 2 hours, 45 minutes, and 15 minutes in cases 3, 4, and 5, respectively. This indicates the robustness of CERRA-1d1kmMYFP in accurately capturing ramp timing and intensity for cases 4 and 5. In case 3, while the timing is simulated with a 2-hour lead, the ramp intensity is closely represented compared to other configurations. Other simulations exhibit larger temporal shifts in cases 4 and 5, and underpredicted intensities in cases 3 and 4. Nonetheless, these findings corroborate the robustness of the WRF model in simulating the extreme ramp events associated with the FLLJ, and the better predictability of the CERRA-1d1kmMYFP modeling configuration.

*Table 2: Overview of the ramp in wind power in terms of intensity (%) and timing (in parenthesis), at a 1-hour time-scale, obtained from the wind power produced by the Belgium offshore wind farm and various CERRA configurations, for cases 3, 4, and 5.*

| Model | Case 3 | Case 4 | Case 5 |
|---|---|---|---|
| **Grid-measured** | 67.0 (09:15) | 52.9 (22:00) | 52.7 (06:30) |
| **CERRA-2d1kmMYFP** | 35.2 (06:00) | 53.3 (20:45) | 55.8 (05:15) |
| **CERRA-2d1kmMYnoFP** | 47.1 (08:30) | 33.4 (20:45) | 57.6 (05:30) |
| **CERRA-2d1kmSH** | 52.9 (08:30) | 29.3 (20:30) | 51.7 (06:00) |
| **CERRA-1d1kmMYFP** | 54.2 (07:15) | 49.9 (21:15) | 58.4 (06:15) |

[Figure]

Figure 6: A comparison of wind power time series from the CERRA-2d1kmMYFP, CERRA-2d1kmMYnoFP, CERRA-2d1kmSH, and CERRA-1d1kmMYFP configurations and the Belgium offshore wind farm production. (a): during 0100 UTC to 1200 UTC on 9th of February, 2016, for case 3. (b): during 1800 UTC on 9th to 0600 UTC on 10th of January, 2017, for case 4. (d): during 0100 UTC to 1200 UTC on 30th of January, 2017, for case 5.

**Specific comments**

1. Line 42-43. Hasn't this weather event been forested by the weather forecast service? Any potential reason for the difficulty? – I mainly want to confirm the down ramp event was not forecasted due to inaccurate weather forecast.

   According to Drew et al. (2018), the National Grid has not utilized advanced weather prediction models from the United Kingdom's Met Office, such as the UK Met Office deterministic model or the UK Met Office Global and Regional Ensemble Prediction System

(MOGREPS). Additionally, no detailed information exists on the specific wind power forecasting methods employed by the National Grid.

2. Line 72-74. A synoptic-scale phenomenon can have features at different scales. The synoptic-scale feature of FLLJ is expected to be better modeled than its local-scale feature such as the timing of passing a wind farm and the intensity at a single location.

Thank you for the suggestion. In the revised manuscript, the following modification is made at P3L71-75, to incorporate the suggestion.

> The FLLJ, though being primarily a synoptic-scale phenomenon, also consists of local-scale features along with their intensity and temporal evolution in specific regions. Thus, while synoptic-scale characteristics are expected to be modeled with relative accuracy, the local-scale features exhibit significant sensitivity to model physics parameterizations, domain configuration, and the resolution of the forcing data.

3. Line 145. What does the criteria refer to?

In the context, the criterion refers to "extreme ramps in wind farm measured power output coincided with the passage of cold fronts in the weather maps."

4. Figure 1 and others. Please add a label to each sub-panel.

Throughout the manuscript, the figures have been updated with sub-panel labels.

5. Line 208-210. What is the main difference between the MYNN2.5 and SH schemes? A brief explanation should help readers understand the rationale behind selecting these schemes and the distinctions they can expect.

The authors thank the reviewer for the suggestion. A detailed comparison and differences between the two PBL schemes is presented in the revised manuscript, at P11L223-237. For reference, the text is presented below.

> The MYNN2.5 and SH PBL schemes differ fundamentally in their approach to turbulence and mixing. MYNN2.5, as a local scheme, relies solely on resolved variables from adjacent grid points to calculate turbulence. In contrast, the SH scheme is nonlocal, using information from multiple vertical levels to determine mixing. It employs scale-aware adjustments and mixing approaches, such as counter-gradient terms or grid-size dependency, to better capture the interplay between the boundary layer and the free atmosphere.

6. Line 217. It appears to be a typo, yet this sentence stands as its own paragraph.

The paragraph break has been eliminated in the revised manuscript.

7. Line 219-224. Please give a reference for the schemes listed here. Change meter to m and kilometers to km.

In the revised manuscript, the appropriate references have been provided to the parameterization schemes. Also, changes meters to m and kilometers to km.

8. Line 249-250. Are the percentage numbers calculated using eq. 1?

Yes, the percentage drop in wind power output measured by the grid are computed using Equation 1.

9. Line 290-291. Please see my preview comment, could you add a label to each panel, so that it can be cited by labels.

Throughout the manuscript, the subfigure labels have been added.

10. Line 315-316. Does the second ramp read from Figure 5?

Yes, a secondary ramp seen CERRA-1d1kmMYFP simulation read from Figure 5(f).

11. Line 421. Does the Elia day-ahead forecast incorporate outputs from numerical weather forecast models? While detailed information isn't available, a high-level introduction would be helpful. Out of curiosity, what does the GFS original time series look like? This insight could be useful in determining whether the GFS serves as a low-cost alternative to Elia and whether dynamic downscaling is necessary for informing farm owners about upcoming down-ramps.

We were unable to find any relevant information on the web. However, we found that the Royal Institute of Belgium created a model to forecast wind power a few years ago. For the wind power forecasts, the Royal Meteorological Institute of Belgium has developed a wind power forecasting system, combining high-resolution deterministic forecasts from the ALARO model, running at a spatial resolution of 4 km and temporal resolution of 15 minutes, and probabilistic ensemble forecasts (ENS) from ECMWF at 18 km resolution and 1-hour intervals. By utilizing ALARO forecasts at turbine hub height and ENS forecasts at 100 m above ground level, the system computes wind power output for wind farms at a 15-minute temporal resolution (Smet et al., 2019).

Regarding the potential of GFS as a low-cost alternative for identifying upcoming down-ramps, we analyzed wind speeds from GFS at the 975 hPa pressure level and compared them with lidar observations. While GFS generally captures the overall trend of observed wind

speeds, it consistently overestimates their magnitude. This overestimation is likely due to GFS's coarse spatial resolution of 0.25°. With such coarse spatial scales, the entire wind farm fits within a single GFS grid cell, which significantly limits its ability to represent the wind power output of the 182 turbines operating within the wind farm. The heterogeneity in turbine locations and wind conditions, which are critical for accurately modeling power output, cannot be resolved at this level of granularity.

[Figure]

*Figure 7: A comparison of 100 m wind speed time series from lidar observations, ERA5, and CERRA reanalysis. The GFS wind speed time series is extracted at 975 hPa level. Left panel: during 0100 UTC to 1200 UTC on 21st of February, 2016, for case 1. Right panel: during 0200 UTC to 1300 UTC on 4th of March, 2016, for case 2.*

Smet, G., Van den Bergh, J., & Termonia, P. (2019, January). Probabilistic storm forecasts for wind farms in the North Sea. ALADIN-HIRLAM NL12.

---

## Referee Report (RR1)

Dear Authors, thank you for your detailed responses to my original comments. You have made many improvements and clarified most of my questions. However, some issues remain, which should be revised before final publication.

My comments to your "Author responses" are given below in blue

**Comments to Author responses**

1. P4L119-120: It's not clear to me what "it's" refers to here
   In the sentence "The jet's acceleration is driven by the isallobaric term, with Coriolis torque and advective tendency terms contributing to its propagation perpendicular to the FLLJ.", it refers to the FLLJ itself. However, the additional terms "perpendicular to the FLLJ" mislead the reader. It should be "parallel to the frontal surface". The entire sentence has been rephrased as below for clarity, and is modified in the revised, at P4L119-120.
   ***The jet's acceleration is driven by the isallobaric term, while the Coriolis torque and advective tendency terms influence its propagation along the frontal surface and adjust its alignment relative to the frontal boundary***
   Excellent, it's much clearer now.

2. P5L145: Why did you choose the two cases from the five as you did? What criteria did you use? The choice of cases is arbitrary. We selected two cases for a complete analysis while the rest of three cases for selected analysis
   Thank you for the clarification; it should be mentioned in the paper also.

3. P6L162: I assume the "forecast" experiment is using GFS? Perhaps it's obvious, but consider stating it explicitly here already
   In the revised manuscript, the forecast experiment is explicitly mentioned in the methods section as presented below.

   At P8L161-163:
   ***For the sensitivity analysis, we chose to vary the initial and boundary conditions, using ERA5 and CERRA datasets to represent hindcast experiments and GFS (Global Forecasting System) to represent the forecast experiment.***

   At P10L195-201:
   ***The initial and boundary conditions provided by reanalysis datasets (e.g., ERA5 and CERRA) tend to be very accurate due to extensive data assimilation. However, in the context of real-time forecasting, such high-fidelity boundary conditions are not available, thus operational forecast data from global models (e.g., GFS) can be used. In the forecast experiment GFS-3d1kmMYFP, cases 1 and 2 are forecasted using a similar configuration as ERA5-3d1kmMYFP, except with GFS real-time forecast data provided as initial and boundary conditions. The GFS IC/BCs during the simulation period are available at a three-hourly resolution and a horizontal***

*grid spacing of about 30km.*

At P11L243-244:
*The GFS IC/BCs initialized at 1200 UTC on 21st February 2016 for case 1 and at 1200 UTC on 3rd March 2016 for case 2 are obtained for the forecast experiment.*

Great. That clears it up.

4. MYNN2.5: what "bl_mynn_*" settings were used? The defaults of WRF v4.4? MYNN2.5 can be quite different depending on this.
   In our simulations we used the following settings for bl_mynn. bl_mynn_tkebudget = 1, bl_mynn_tkeadvect = .true., On the other hand, the other settings are unaltered, thus taken as their default settings.
   Thank you for the clarification. You should describe at least these non-default settings in the paper to allow others to reproduce the results.

5. In a study focussing on capturing the timing of an event, it's surprising to me that you don't consider the influence of data assimilation (except to say that is one reason for the accuracy of re-analysis datasets). Why did you not test data assimilation and/or discuss this in the paper?
   It is true that data assimilation can be a crucial factor, especially when it comes to real-time forecasting, where accurately initializing the model states is key to improving short-term predictions. However, in this study, our main objective was to examine the sensitivity of modeling parameters and assess the ability of the WRF model to simulate frontal low-level jets and associated extreme ramp events. By focusing on the modeling aspects, we aim to better understand the dynamics and physical processes involved in the event simulation, separate from the impacts of assimilation techniques.
   Your paper strongly emphasizes modeling the timing of ramp events, so data assimilation remains a relevant missing factor. I will maintain that you should do more to discuss the possible consequences of not including DA here as it relates to the conclusions about choosing your nesting and model domains.

6. You use 51 levels, why not more? Are you sure it's not sensitive to this?
   In the revised manuscript, we have examined the influence of vertical levels on the simulations by selecting 101 vertical levels, with approximately 27 vertical levels within the first 1 km of height. Using this configuration and the CERRA-1km1dMYFP simulation strategy, the first two cases were simulated. From the analysis, it was found that the vertical levels exert noticeable influence on the ramp timing and post-ramp wind speed. However, the performance of the 51 vertical levels configuration was better than that of 101 vertical levels, further justifying our choice of vertical levels. The analysis is presented in Appendix B in the revised manuscript and is presented below for reference.
   *We have examined the influence of vertical levels on the simulations by selecting 100 vertical levels, with approximately 27 vertical levels within the first 1 km of height. Using this vertical resolution and the CERRA-1km1dMYFP configuration,*

*the first two cases were simulated. In our study, the choice of 50 vertical levels was adopted from Nunalee and Basu (2014), where in the case of coastal LLJs, the authors reported reduced jet strength and lower jet core height with increased vertical levels, deviated from the observations. Similar to their findings, we noticed the vertical levels exert considerable influence on the ramp timing and marginal influence on the ramp intensity, as shown in Fig. B1. However, we also recognize the limitations of our findings, since they are based on two simulations focused specifically on FLLJ cases.*

This is a nice addition to the paper, providing clear evidence that adding more levels matters, but not with a clear positive on your application.

7. You used 6 hr of spin-up. Are you sure the model has enough time to develop these strong weather events? Perhaps the poor performance of some of the experiments is due to insufficient spin-up time. See e.g. Lui et al. (2023)

To evaluate the sensitivity with respect to spin-up time, we conducted simulations for the two cases using the CERRA-1d1kmMYFP configuration, one with a 6-hour spin-up and another with a 12-hour spin-up. The results revealed that the wind power output from both simulations was identical, indicating that a 6-hour spin-up was sufficient for the development of the FLLJ cases. This confirms the adequacy of the chosen spin-up duration for accurately capturing the dynamics of these events. The following analysis is presented in Appendix C in the revised manuscript.

*To evaluate the sensitivity of model accuracy to the choice of spin-up time, we conducted simulations for two cases using the CERRA-1d1kmMYFP configuration: one with a 6-hour spin-up (original simulation duration) and another with a 12-hour spin-up (starting 6 hours prior to the original duration). Figure C1 presents wind power time-series from both simulations, compared with the measured power output for Cases 1 and 2. The results show that the wind power output from both simulations is identical, indicating that a 6-hour spin-up is sufficient for the development of the FLLJ cases. However, it is highly unlikely for simulations initialized with different IC/BCs to produce identical results. To further verify this, we compared the time-height crosssection of wind speed from the two simulations, along with lidar observations at the LOT2 location, for Cases 1 and 2, as shown in Fig. C2. The comparison reveals that the simulations are nearly identical, with no discernible differences in wind speed, except for minor variations marked by sky-blue circles in both cases. Since no differences are observed in wind speed below the 100 m level, the wind power outputs from the two simulations also remain identical, as shown in Fig. C1. These findings confirm that the chosen 6-hour spin-up duration is adequate for accurately capturing the dynamics of these events.*

Save as above. This is a helpful addition to the paper, ruling out influences of spin-up time.

8. Figures 4-10: Why not show the results from the datasets used for forcing data: ERA5, CERRA, and GFS? It would be more convincing to show that downscaling is needed if I

We appreciate the reviewer's suggestion to include the results from the forcing datasets (ERA5, CERRA, and GFS) in Figures 4–10. However, we are unable to include these results in the manuscript for the following reasons. We did analyze wind speeds from ERA5 and CERRA at 100 m and from GFS at the 975 hPa pressure level, comparing them with lidar observations, which are shown in Fig. D1, and are presented here for reference. Our analysis revealed that while GFS follows the overall trend of observed wind speeds, it exhibits a clear overestimation in magnitude. Though the GFS data at the current time exist at an hourly temporal resolution, it is only available at a 3- hourly resolution for 2016-17 years. Due to the coarse temporal resolution, it is challenging to quantify ramp statistics, particularly as extreme ramps associated with frontal low-level jets occur on scales of minutes. This further demonstrates that dynamical downscaling is indispensable for understanding extreme wind ramps at sub-hourly scales, which are critical for wind power applications

Furthermore, and most importantly, CERRA and ERA5 provide wind speeds at 100 m above ground level, while GFS provides wind speeds at the 975 hPa pressure level. However, four of the five wind turbine types operating in the wind farm have hub heights below 100 m, making it challenging to accurately approximate wind power production using the forcing datasets.

Additionally, most of the analysis from WRF simulations in this study was conducted at temporal scales of 10–15 minutes, whereas the temporal resolutions of the forcing datasets range from 1 to 3 hours. This mismatch in temporal resolution further limits the utility of the forcing data in capturing rapid changes and short-term wind fluctuations associated with extreme ramps.

The ERA5 reanalysis exists at a spatial resolution of 0.25°, the CERRA analysis exists at a spatial resolution of 5.5 km, and the GFS forecast exists at a spatial resolution of 0.25°. Due to these coarse spatial resolutions, the entire wind farm fits within a single grid cell for ERA5 and GFS, while only a few grid cells from CERRA cover the wind farm. These spatial limitations hinder the representation of the wind power output generated by 182 turbines within the wind farm. This is critical, as the heterogeneity in turbine locations and wind conditions cannot be resolved with the coarser spatial scales of the forcing data.

Including the analysis from the forcing datasets would also increase the number of outputs by three, significantly complicating the visual presentation of Figures 4–10. We believe excluding the forcing data maintains the clarity and focus of the manuscript while strengthening the argument for the necessity of dynamical downscaling to capture subhourly wind variability. The justification regarding the necessity of dynamical downscaling is incorporated at PL in the revised manuscript

I agree that the forcing datasets are probably less appropriate for extreme wind ramps due to the reasons you list. However, that is precisely why it would be valuable to show the significant improvements from your dynamical downscaling relative to the forcing data. Both ERA5 and CERRA are available hourly at many levels. See, e.g., https://cds.climate.copernicus.eu/datasets/reanalysis-cerra-height-levels?tab=download, and

9. The range of values in the colormap makes some of the plots difficult to it difficult to interpret. For example, do you really need to include values up to 30 m/s in figures D1-4. In Fig. C1, the breaks in the colormap seem to be inconsistent (sometimes 2 m/s, sometimes 1 m/s)

In the revised manuscript, the color levels of Figures F1-4 are set according to the minimum and maximum of the wind speed, at an interval of 1 m/s. For Fig. E1, the color levels are set from 4 to 14 m/s at an interval of 2 m/s and from 15 to 28 m/s at an interval of 1 m/s. This gives clarity in visualizing the wind speed contours for cases 1 and 2.

The new levels help, but I recommend choosing a different colormap without so many discontinuities/breaks. See, e.g., https://www.fabiocrameri.ch/colourmaps/.

10. Do you use the ERA5 pressure-levels or model-levels?

We have used the ERA5 pressure level data as initial and boundary conditions.

Thank you for the clarification here. You should also clarify this in the paper.

11. In P14L298-300: What exactly are you arguing here? that WRF cannot generate sharp gradients from coarse ERA5 boundary-condition data? You say CERRA constitutes better BCs, but it is itself based on ERA5.

In the referenced passage, we argue that while CERRA is derived from ERA5, it incorporates additional data assimilation and a significantly higher horizontal resolution (5.5 km vs. 31 km), which enhances its representation of mesoscale features. This improved resolution and assimilation process allow CERRA to better capture sharp gradients, such as those associated with the Frontal Low-Level Jet, providing more accurate boundary conditions for WRF simulations.

So, one way to see the results is that, since you don't include data assimilation, the best option is to rely mainly on the high-resolution CERRA IC/BCs to obtain good features and timing by using only 1 WRF domain and reducing the fetch from the boundary to the region of interest. Your innermost domain is centered on what looks like approx. 30 km northeast of the wind farms in the direction opposite to the prevailing wind direction, leaving approx. 120 km of fetch for the prevailing wind direction.

12. P23L423: Why are you surprised that the model captured the event? Please elaborate on why it's surprising

The intention was not to express surprise at the WRF model's ability to forecast the ramp and extremity of the FLLJ but rather to highlight the contrasting performance of the Elia forecast. Specifically, while the WRF model captured the ramp events in both cases, albeit with some timing mismatch, the Elia forecast either predicted a gradual ramp-down (case 1) or did not capture the ramp at all (case 2). This discrepancy was noteworthy because it suggests potential challenges in operational forecasting of such extreme

events. We have revised the text to clarify this point and to better reflect the intended message without any exaggeration. The revised manuscript is updated with the following text, at P22L443-444.

***We note that the WRF model was able to forecast the extreme ramp event in both cases, though with some degree of timing mismatch.***

Thank you for the clarification.

13. Figure 11: Unfortunately, only the CERRA-1d1kmMYFP was included here, it would have been more convincing to show that the trend from cases 1 and 2 continues here

We appreciate your interest in extending the comparison to demonstrate the trends from Cases 1 and 2 within this figure. The three additional cases have been simulated using the CERRA-based configurations, namely CERRA-2d1kmMYFP, CERRA-2d1kmMYnoFP, CERRA-2d1kmSH, and CERRA-1d1kmMYFP. The following analysis, figure, and table are incorporated in the revised manuscript, at P26L463-476.

***The results demonstrate that the WRF model successfully simulates the extreme wind power ramps associated with FLLJs, though the specific modeling configurations significantly influence the wind power time series. All simulations capture the strong pre-ramp wind power at the 720 MW rated capacity, likely due to peak wind speeds during the FLLJ, as well as the post-ramp wind power output, which aligns with observed values. The observed strong drop in wind power, representing the extreme ramp, is consistently simulated across cases; however, discrepancies in ramp intensity and timing persist, varying across configurations. The ramp statistics from the four CERRA-based configurations for three cases are computed at a 1-hour time scale and are presented in Table 4. The CERRA-1d1kmMYFP simulation shows superior performance, with the power ramps surpassing 50% within just 1 hour, signifying the extremity of the power ramps. In terms of ramp timing, the simulated ramps are in advance by 2 hours, 45 minutes, and 15 minutes in cases 3, 4, and 5, respectively. This indicates the robustness of CERRA-1d1kmMYFP in accurately capturing ramp timing and intensity for cases 4 and 5. In case 3, while the timing is simulated with a 2-hour lead, the ramp intensity is closely represented compared to other configurations. Other simulations exhibit larger temporal shifts in cases 4 and 5, and underpredicted intensities in cases 3 and 4. Nonetheless, these findings corroborate the robustness of the WRF model in simulating the extreme ramp events associated with the FLLJ, and the better predictability of the CERRA-1d1kmMYFP modeling configuration.***

It's highly appreciated that you extended your analysis to the three remaining cases. The trend appears reasonably consistent, with CERRA-1d1kmMYFP performing best (at least for 4 out of 5 events). Perhaps you could highlight (bold text, perhaps?) the best-performing simulation in Tables 2, 3, and 4 for intensity and timing.

14. P27L463: Same as above, you say CERRA provides better BCs than ERA5 to capture timing and intensity, but CERRA is based on downscaling from ERA5, so perhaps the problem is not ERA5 BCs but the downscaling and e.g. lack of data assimilation?

In continuation, as stated in response 5, while CERRA is based on downscaling from ERA5, it provides better boundary conditions (BCs) due to its higher resolution (5.5 km) and the frequent incorporation of observational data through data assimilation techniques on a regional scale. These factors enhance CERRA's ability to represent mesoscale processes, allowing for more accurate reproduction of the timing and intensity of atmospheric phenomena like frontal low-level jets. That said, we acknowledge that data assimilation could significantly impact the results, particularly for real-time simulations and improving the initial state of the atmosphere. However, data assimilation is out of scope for the current study, since many different approaches exist, such as 3DVAR, 4DVAR, Kalman filter, which perform differently as per the literature. In this study, our primary focus was on evaluating the model's sensitivity towards different modeling configurations and examining the differences between using higher and lower resolution IC/BCs

See my answer under 5)

**Additional comments**

- Please complete the "data availability" section related to all the data used and produced in your study. What about the availability of your simulations, the LiDAR data, and so on?
- Consider providing more detail in the "author contribution" section. See, e.g., https://publications.copernicus.org/services/contributor_roles_taxonomy.html

---

## Author Response (AR2)

Dear Reviewer, thank you for your thorough feedback on our revised manuscript. Your comments have further improved the quality and readability of our manuscript.

In response to your comments, we have incorporated some new insights and modified the revised manuscript further. Specifically, we have incorporated the ramp statistics from ERA5 and CERRA wind speed data, along with a paragraph detailing the consequences of not incorporating the data assimilation into the conclusions.

Our responses to your comments are given below. We have only included the comments that are unanswered or needed revision.

**Major comments**

2. Thank you for the clarification; it should be mentioned in the paper also.

   The choice of the cases is mentioned in the revised manuscript at P5L147-148 and is provided here for reference.

   > The choice of the two cases for primary focus are arbitrary.

4. Thank you for the clarification. You should describe at least these non-default settings in the paper to allow others to reproduce the results.

   The MYNN PBL settings are provided at P11L239-240, in the revised manuscript, and is provided here for reference.

   > For the simulations with MYNN2.5 PBL scheme, we set bl_mynn_tkebudget = 1 and bl_mynn_tkeadvect. = true for the bl_mynn_settings, while the other settings are unaltered, taking their default settings.

5. Your paper strongly emphasizes modeling the timing of ramp events, so data assimilation remains a relevant missing factor. I will maintain that you should do more to discuss the possible consequences of not including DA here as it relates to the conclusions about choosing your nesting and model domains.

   We acknowledge the reviewer's comment that data assimilation plays a key role in short-term weather forecasting. While our study mainly focusses on modelling aspects to better understand the dynamics and physical processes involved in the extreme-ramp event simulations, we would extend our study to examine the influence of data assimilation on extreme-ramp forecasting in the future.

   In the revised manuscript, we have incorporated the following text at P29L521-535.

   > Data assimilation (DA) is a crucial factor in numerical weather prediction, particularly for real-time forecasting, where accurate initialization of model states enhances short-term predictability. As demonstrated in the studies by Wilczak et al. (2015, 2019), DA has been shown to significantly improve short-term wind power forecasts by reducing forecast biases

and enhancing the representation of wind ramps. Specifically, the assimilation of in situ and remote sensing observations into high-resolution models has led to up to a 6\% reduction in RMSE for short-term wind power predictions, with notable improvements in the first few forecast hours. However, the effectiveness of DA is highly dependent on factors such as the type of observations assimilated, the assimilation technique employed, and the forecast lead time. Ivanova et al. (2025) demonstrated the benefits of Four-Dimensional Data Assimilation (FDDA) of lidar observations in improving wind speed and wind power forecasts for Belgian offshore wind farms. However, they also identified key limitations, including the dependence on specific wind directions, the scarcity of real-time offshore observations, and the restriction of FDDA to prognostic variables. While these findings reinforce the potential benefits of DA, our study had a different objective. Rather than focusing on assimilation techniques, we aimed to assess the sensitivity of modeling parameters and evaluate the WRF model's ability to simulate frontal low-level jets and extreme ramp events. In addition, we sought to improve our understanding of the underlying atmospheric processes. Future work could explore the added advantage of DA and assess its impact on extreme wind events at the resolutions and configurations considered here.

Wilczak, J., Finley, C., Freedman, J., Cline, J., Bianco, L., Olson, J., Djalalova, I., Sheridan, L., Ahlstrom, M., Manobianco, J. and Zack, J., 2015. The Wind Forecast Improvement Project (WFIP): A public–private partnership addressing wind energy forecast needs. *Bulletin of the American Meteorological Society*, *96*(10), pp.1699-1718.

Wilczak, J.M., Olson, J.B., Djalalova, I., Bianco, L., Berg, L.K., Shaw, W.J., Coulter, R.L., Eckman, R.M., Freedman, J., Finley, C. and Cline, J., 2019. Data assimilation impact of in situ and remote sensing meteorological observations on wind power forecasts during the first W ind F orecast I mprovement P roject (WFIP). *Wind Energy*, *22*(7), pp.932-944.

Ivanova, T., Porchetta, S., Buckingham, S., Glabeke, G., van Beeck, J. and Munters, W., 2025. Improving wind and power predictions via four-dimensional data assimilation in the WRF model: case study of storms in February 2022 at Belgian offshore wind farms. *Wind Energy Science*, *10*(1), pp.245-268.

8. I agree that the forcing datasets are probably less appropriate for extreme wind ramps due to the reasons you list. However, that is precisely why it would be valuable to show the significant improvements from your dynamical downscaling relative to the forcing data. Both ERA5 and CERRA are available hourly at many levels. See, e.g., https://cds.climate.copernicus.eu/datasets/reanalysis-cerra-height-levels?tab=download, and https://cds.climate.copernicus.eu/datasets/reanalysis-era5-complete?tab=d_download. I agree with you that adding more datasets to the figures could lead to cluttering. However, you could add them to Tables 2 and 4, which would indicate the improvements from downscaling.

We thank the reviewer for the suggestion. Accordingly, we included the ramp statistics from CERRA and ERA5 wind speeds available at different model levels. In the revised manuscript, the statistics are added in Tables 2 and 4, while the following text is included at P14 L280-285 and P26L488-490.

To highlight the necessity of dynamical downscaling, wind power output is computed using wind speed from ERA5 model levels (https://cds.climate.copernicus.eu/datasets/reanalysis-era5-complete?tab=d_download) and CERRA height levels (https://cds.climate.copernicus.eu/datasets/reanalysis-cerra-height-levels?tab=download). The corresponding ramp statistics are presented in Table \ref{ramp_in_capacity_factor_combined}. The results indicate that ramp timings derived from both datasets closely align with those from WRF simulations and observations. However, the ramp intensity is significantly underestimated by ERA5, whereas CERRA exhibits a moderate underestimation.

Similar to Cases 1 and 2, both ERA5 and CERRA significantly underestimate the ramp intensity in these additional cases, with CERRA exhibiting an overestimation in Case 5. However, the ramp timings remain comparable to those obtained from the WRF simulations.

9.  The new levels help, but I recommend choosing a different colormap without so many discontinuities/breaks. See, e.g., https://www.fabiocrameri.ch/colourmaps/.

    In the revised manuscript, Figures E1 and F1-4 are redraw with suggested colour map, without having many discontinuities.

10. Thank you for the clarification here. You should also clarify this in the paper.

    In the revised manuscript, a clarification is provided at P10L194 and L196 and is provided here for reference.
    We have used the pressure level data set from both ERA5 and CERRA as initial and boundary conditions in our simulations.

13. It's highly appreciated that you extended your analysis to the three remaining cases. The trend appears reasonably consistent, with CERRA-1d1kmMYFP performing best (at least for 4 out of 5 events). Perhaps you could highlight (bold text, perhaps?) the best-performing simulation in Tables 2, 3, and 4 for intensity and timing.

    In the revised manuscript, the best performing simulations are underlined in Tables 2, 3, and 4.

14  See my answer under 5)

    As stated in response 5, we have incorporated a detailed text regarding consequences of not using data assimilation and a scope for future work.

**Additional comments**

1.  Please complete the "data availability" section related to all the data used and produced in your study. What about the availability of your simulations, the LiDAR data, and so on?

    In the revised manuscript, an elaborated "data availability" section is provided, explaining how to obtain the simulated data and observations. For reference, the text is provided here.

    The datasets used in this study are publicly available. The ERA5 and CERRA reanalysis are downloaded from ECMWF CDO, available at

https://cds.climate.copernicus.eu/cdsapp\#!/search?type=dataset. The scripts and workflows for data processing and analysis can be accessed in the GitHub repository https://github.com/HarishBaki/Modeling-Frontal-Low-Level-Jets-and-Associated-Extreme-Wind-Ramps.git. The wind model simulations and observational datasets supporting this study are hosted on Zenodo, at https://doi.org/10.5281/zenodo.15033463. These repositories ensure transparency and reproducibility, allowing further exploration and validation of the study's findings.

2    Consider providing more detail in the "author contribution" section. See, e.g., https://publications.copernicus.org/services/contributor_roles_taxonomy.html

The author contribution section is modified according to the journal standards. For reference, the text is provided here.

HB is responsible for Conceptualization, Data curation, Formal analysis, Investigation, Methodology, Visualization, and Writing – original draft preparation. SB is responsible for Conceptualization, Supervision and Writing – review & editing. GL contributed Supervision and Writing – review & editing.